# Fe-rich X-ray amorphous material records past climate and persistence of water on Mars
Anthony D. Feldman [1,5] ✉, Elisabeth M. Hausrath [1] ✉, Elizabeth B. Rampe [2], Valerie Tu[3], Tanya S. Peretyazhko[3], Christopher DeFelice [1,6] & Thomas Sharp [4]

X-ray amorphous material comprises 15-73 wt.% of sedimentary rocks and eolian sediments in Gale crater. This material is variably siliceous and iron rich but aluminum poor. The presence of volatiles is consistent with the existence of incipient weathering products. To better understand the implications of this material for past aqueous conditions on Mars, here we investigate X-ray amorphous material formation and longevity within terrestrial iron rich soils with varying ages and environmental conditions using bulk and selective dissolution methods, X-ray diffraction, and transmission electron microscopy. Results indicate that in situ aqueous alteration is required to concentrate iron into clay-size fraction material. Cooler climates promote the formation and persistence of X-ray amorphous material whereas warmer climates promote the formation of crystalline secondary phases. Iron rich X-ray amorphous material formation and persistence on Mars are therefore consistent with past cool and relatively wet environments followed by long-term cold and dry conditions.

X-ray diffraction (XRD) measurements by the Chemistry and Mineralogy (CheMin) instrument on the *Curiosity* rover at Gale crater, Mars have demonstrated that X-ray amorphous material makes up 15% to 73 wt% of all rock and eolian sediment samples analyzed to date[1–3]. This material has been shown to be variably Fe-rich (3.0–43.1 wt% $FeO_T$) and Si-rich (3.6–75.9 wt% $SiO_2$) and typically Al-poor (≤10.46 wt% $Al_2O_3$)[1–4], consistent with principally mafic sources[5,6]. This material is hereafter referred to as "amorphous" although it likely encompasses amorphous and short-range order materials and nanocrystallites[7–10]. Despite the prevalence of amorphous material in Gale crater[1–4] and elsewhere on Mars[11–22], the implications of its presence for past environmental conditions on Mars remain poorly understood[4]. The Gale crater amorphous component could represent in-situ pedogenic or diagenetic alteration products, chemical precipitates, or detrital influx of glasses from dust or fluvial inputs[4]. While amorphous material can form from a variety of processes, including as glass from melt and through hydrothermal alteration processes[23,24], the presence of volatiles (e.g., $H_2O$, $CO_2$, $SO_2$) measured by the Sample Analysis at Mars (SAM) instrument suite suggests at least some of this material likely consists of secondary weathering products[1,4,25,26]. As amorphous material represents a substantial component of the Gale Crater sedimentary rocks and eolian sediments and is present across the Martian surface, elucidating the processes controlling the formation and preservation of this material is critical for understanding sedimentation and diagenetic processes on Mars[4,27].

The elevated abundances of amorphous material in Noachian to early Hesperian sediments on Mars was unexpected as amorphous material is often understood to represent metastable material susceptible to conversion to more thermodynamically stable and more crystalline mineral phases[28,29]. Previous work on terrestrial soils has identified amorphous aluminosilicates as precursors to crystalline clay minerals[29–35] and suggested aluminous amorphous material preferentially forms through aqueous alteration under kinetically limiting conditions[29,36] such as colder temperatures[35–39]. Variably Fe-rich (0.95–45.17 wt% $FeO_T$) and Si-rich (0.23–93.26 wt% $SiO_2$) but Al-containing (0.81–55.79 wt% $Al_2O_3$) amorphous material has recently been documented in terrestrial glacial sediments[9,35], as well as in Hawaiian soils and paleosols at the John Day Fossil Beds[9]. However, little work has yet examined climatic effects on the formation and persistence of Al-poor yet Fe-rich amorphous material as is common in Gale crater sediments and rocks. Such data are crucial for placing constraints on environmental conditions present throughout the formation and persistence of chemically similar amorphous material on Mars.

In this study, we use chemical dissolution techniques coupled with powder XRD and transmission electron microscopy (TEM) to investigate

[1]Department of Geosciences, University of Nevada Las Vegas, Las Vegas, NV, USA. [2]Astromaterials Research and Exploration Science Division, NASA Johnson Space Center, Houston, TX, USA. [3]Jacobs Technology, NASA Johnson Space Center, Houston, TX, USA. [4]School of Earth and Space Exploration, Arizona State University, Tempe, AZ, USA. [5]Present address: Desert Research Institute, Las Vegas, NV, USA[6]Present address: Pacific Northwest National Laboratory, Richland, WA, USA. ✉e-mail: anthony.feldman@dri.edu; elisabeth.hausrath@unlv.edu

the impacts of time and climate on the formation and persistence of Fe-containing but Al-poor amorphous material. We examined the formation and persistence of amorphous Fe-containing material in ultramafic soils developing in the Mediterranean climate Klamath Mountains of northern California[40–43], the subarctic climate Tablelands of Newfoundland, Canada[44], and the desert climate of Pickhandle Gulch, Nevada[40,41], with soils spanning in age from ~12 to >50 ka (Fig. S1). Our results show that cool and wet conditions lead to greater in situ formation and persistence of Fe-rich secondary amorphous material. The presence of abundant Fe-rich amorphous material at Gale Crater is therefore consistent with cool and wet conditions during their formation, followed by cold and dry conditions promoting their persistence.

## Field site selection

Serpentine soils provide a useful analog setting to examine processes in Fe-rich and Al-poor settings that are chemically relevant to the amorphous material and secondary minerals present in Gale Crater. Soils derived from such hydrothermally altered ultramafic rock are typically Mg/Fe/Si-rich and Al-poor[42–46], chemically similar to the Al-poor and variably Fe/Si-rich chemical composition of the Gale crater X-ray amorphous component[1,2]. Fe-oxides and smectite clay minerals comprise the majority of secondary products in serpentine soils[43,47–55]. Likely secondary amorphous material has also previously been postulated in such settings[51]. As a result of these chemical and mineralogical similarities to many Martian materials, serpentine laterites[52] and serpentinite rock[23,56,57] have previously been investigated as analog environments for aqueous alteration processes in Fe-rich and Al-poor martian systems. Partially serpentinized lake sediments have similarly been utilized as analog environments for analyzing alteration patterns in Martian paleolake settings with primarily mafic mineral sources[58,59].

Four serpentinite bodies in three different climate zones were chosen for investigation to examine climatic influences on the production and persistence of Fe-containing amorphous material. These sites include the Trinity Ultramafic Body and Rattlesnake Creek Terrane within the Klamath Mountains of northern California, the Tablelands of Gros Morne National Park in Newfoundland, Canada, and a serpentinite body near the ghost town of Pickhandle Gulch in western Nevada (Table S1, Fig. S1). The modern climate in the Klamath Mountains is Mediterranean, featuring hot and dry summers and cold and wet winters. NOAA 1991–2020 climate normals for the Weaverville, CA weather station (Station ID: USC00049490), near the String Bean Creek sampling site in the Rattlesnake Creek Terrane, record mean annual temperatures of 12.8 °C, with a mean of 23.2 °C in July and 3.7 °C in December, and an annual average precipitation of ~100 cm[41,60], predominantly snowfall. NOAA 1981–2010 climate normals for the Sawyers Bar Ranger Station (Station ID: USC00048025), north of Trinity Lake and within the Trinity Alps, show a mean annual temperature of 12.8 °C, with a mean of 22.9 °C in July and 4.4 °C in December, and an annual average precipitation of ~118 cm falling mostly as snow[41,60]; 1991–2020 climate normals were not available for that site. The exact climate at the Eunice Bluff and Deadfall Lake sites in the Trinity region is likely slightly colder due to a higher elevation (~2100 m) compared to ~600–650 m elevation for the monitoring stations. Similarly, high-elevation regions of the Klamath Mountains have mean summertime temperatures (July–October) of ~12.7 °C and mean wintertime (November–May) temperatures hovering around freezing (~0.9 °C), though with substantial periods of below-freezing conditions and abundant snowfall[61]. A dispersed coniferous forest covers much of the Klamath Mountains[60,62] and is present at all sampling locations examined within this study. The modern climate of the Tablelands is subarctic; daily mean temperatures at the nearby Cow Head weather station (Station ID: 8401335) maintained by Environment Canada range from −7.0 °C in January to 15.7 °C in August with a mean yearly temperature of 3.9 °C, with a yearly average precipitation of 120.4 cm distributed relatively evenly throughout the year as both snowfall and rain[63]. The Tablelands are relatively barren, with occasional ground-hugging shrubs and stunted pine trees[64,65]. Pickhandle Gulch possesses a desert climate, with the nearest NOAA climate station to Pickhandle Gulch, in Mina,

Nevada (Station ID: USC00265168), recording 1991–2020 climate normals of a 14.4 °C mean annual temperature, ranging from 2.7 °C in December to 27.9 °C in July, with an average precipitation of 14.2 cm annually[41]. Vegetation at Pickhandle Gulch is a sparse sagebrush cover typical of the Great Basin region[66].

## Results and discussion

### Characteristics of the soil system: soil morphology, pH, and organic content

Although the Klamath Mountains and Tablelands soils possess similar age ranges, soils exhibit greater visual indications of in-situ pedogenic alteration within the Klamath Mountains than at the Tablelands or at Pickhandle Gulch—see soil descriptions and pH and loss on ignition at 550 °C (LOI$_{550}$) values in Supplementary Data 3. As soil parent material and soil ages are similar in the Klamath Mountains and Tablelands, we attribute variations in soil development to climatically induced differences in in-situ aqueous alteration. Here we provide background on the soil environment in which secondary amorphous material is forming.

Klamath Mountain soils exhibit a clear age progression. Munsell color hue varies from 10YR to 7.5YR in the youngest Klamath Mountain soils (Eunice Bluff and Deadfall Lake, ~12.1 ka), with 5YR hues observed in the oldest Klamath Mountain soil (String Bean Creek, undated). These increases in soil reddening are consistent with the formation of ferrihydrite and goethite, and potentially hematite as soil age increases[67,68]. Similarly, clay films are absent from the youngest Klamath Mountain soils but progressively become more visible as soil age increases, with clay films observed throughout the entire soil profile in the oldest Klamath Mountain soil except for in the surface A horizon. The increase in clay-film visibility with soil age is consistent with the formation of secondary clay minerals. Soil peds are likewise absent or present as < 5 mm diameter sub-angular blocky shapes in the youngest Klamath Mountain soils, with sub-angular blocky peds up to 5 cm in diameter present in the oldest Klamath Mountain soil. Klamath Mountain soil textures include loamy sands, sandy loams, and sandy clay loams.

Tablelands soils are relatively uniform in appearance regardless of soil age. Munsell hue varies from 2.5Y in the younger soils (~13–17.6 ka) to exclusively 10YR in the oldest (Trout River Gulch, >20 ka), consistent with the formation of Fe-oxides such as ferrihydrite and goethite[68]. Indications for ped formation or clay-film development were not observed in any Tablelands soil. Tablelands soil textures include loamy sands, sandy loams, and sandy clay loams. Coincident with the relative lack of variation with age, Tablelands soils do not possess visual indications for soil horizon differentiation. Such lack of soil horizon differentiation is potentially attributable to freeze-thaw cycling and frost-heaving mixing soil material and preventing horizon formation, as has been previously observed in soils on the Tablelands[69].

Pickhandle Gulch soils do not exhibit significant variation. Soils are thin (<10 cm), likely reflecting a predominantly detrital buildup with limited contributions from underlying bedrock. All soils exhibit Munsell colors with 10YR hues, a chroma of 4, and values of 2 or 3. Soils either lack ped development or possess weakly developed sub-angular blocky peds < 5 mm in diameter. Clay-films are absent from peds when present. Soil texture encompasses loamy sands, sandy loams, sandy clay loams, and sandy clays. Formation of soil pores at the surface of each soil profile is consistent with the development of vesicular horizons through dust influx and sporadic precipitation[70,71].

Soil pH and LOI$_{550}$, a proxy measure of total organic content, likewise vary between field sites. Soil pH is slightly acidic to slightly alkaline (5.88–7.56) in the Klamath Mountains, slightly alkaline (7.30–8.30) in the Tablelands, and alkaline at Pickhandle Gulch (8.17–8.72). LOI$_{550}$ values (uncertainty estimated as ±2%[72]) are greatest near the surface of Klamath Mountain soils, varying from as much as 21.97 wt% at the surface to between 1.67 and 4.69 wt% at depth. The vertical variation in LOI$_{550}$ with a relatively monotonic decrease with depth likely reflects heterogenous distribution of organics within the soil profile resulting from contributions of

organic litter from the overlying dispersed coniferous forest vegetation. $LOI_{550}$ values in the Tablelands (6.24–11.06 wt%) and at Pickhandle Gulch (1.03–2.32 wt%) are both substantially lower than the maxima observed in the Klamath Mountains and exhibit limited vertical variability, reflecting the lack of vegetation cover at both field sites and a corresponding lack of direct leaf/plant litter contributing to organic enrichment at the soil surface. Variations in organic content are reflected in varying, though limited, organically bound Fe-content as measured by a Na-pyrophosphate extraction[73,74] (Fig. S53; Supplementary Data 2). While $Fe_P$ is minimal in all examined soils (≤1.16 mg Fe/g soil), relatively higher $Fe_P$ in the Klamath Mountains (0.11–1.16 mg Fe/g soil) than Tablelands (0.02–0.30 mg Fe/g soil) or Pickhandle Gulch (0.00–0.03 mg Fe/g soil) soils is consistent with the relative contributions of organics as measured by $LOI_{550}$.

## Soil mineral inputs to the amorphous material-forming weathering system

Secondary material production is inextricably linked to the parent material from which it forms. Collected rock samples interpreted as parent material at the Klamath Mountain sites contain serpentine minerals, along with variable amounts of amphibole, chlorite, enstatite, olivine, plagioclase, and talc (Table 1, Fig. S3). The Tablelands parent material is composed of serpentine minerals, chlorite, magnetite, talc, and olivine (Table 1, Fig. S3), and the parent material at Pickhandle Gulch contains serpentine minerals, chlorite, and magnetite (Table 1, Fig. S3).

Dust input also potentially contributes to the composition of the secondary phases. Dust inputs to the examined soils were assessed by comparing the bulk soil (< 2 mm diameter) and clay-size fraction (< 2 μm diameter) to the parent material mineralogy and chemistry at each location. Minerals were attributed to dust input if they would not have formed in situ through aqueous alteration at surface conditions and were absent from parent material. Relative XRD peak heights indicate minor concentrations of quartz in the Tablelands soil clay-size fractions and of quartz and plagioclase in the Klamath Mountains soil clay-size fractions (Fig. 1A, S4, S6, S8, S10, S12, S14), consistent with relatively minor dust influx at those sites. In contrast, the clay-size fractions at Pickhandle Gulch incorporate abundant phases attributable to dust deposition, including calcite, muscovite, plagioclase, quartz, and a substantial smectite component (Fig. 1A, S24), all common phases in southwestern Nevada dust[75]. High Al-content in Pickhandle Gulch clay-size fraction material relative to parent material (Fig. 2) is similarly consistent with a high concentration of exogenous dust-borne phases.

Examination of the persistence and absence of parent material minerals within each soil can also help understand contributions from primary mineral dissolution to secondary material formation. In these soils, parent material minerals generally persist in the soils formed from those parent materials (Table 1), with three notable exceptions. Magnetite is present in some parent materials but not within examined soils (Table 1), consistent with dissolution or presence below XRD detection limits. Similarly, pyroxene is present within Eunice Bluff parent material but absent from the soil at the Eunice Bluff site (Table 1), indicating complete dissolution within <12.1 ka, corresponding to previous estimates of pyroxene persistence within terrestrial soils[76], and releasing elemental chemistry likely including Mg, Si, and Fe to secondary phase formation. Likewise, olivine is present in the Klamath Mountains at the 12.1 ka Eunice Bluff and Deadfall Lake soils but absent from older (>15 ka) sites despite being present in parent rock samples of all Klamath sites except Deadfall Lake (Table 1). Olivine dissolution releases Mg, Fe, and Si that would be available to secondary phase formation, and the ~12–15 ka persistence time for olivine is roughly equivalent to previous observations of ~10 ka olivine persistence time within terrestrial soils[76].

In contrast, although relative XRD peak height suggests that olivine is less abundant in the Tablelands than in Klamath Mountain parent materials (Fig. S4), olivine is present in all examined bulk soils from the Tablelands to >20 ka but absent from the clay-size fraction (Table 1), indicating only partial dissolution and increased olivine persistence relative to the warmer

**Table 1 | Mineral presence in bulk soil, soil clay-size fraction, and parent material determined by oriented and randomly oriented XRD pattern analysis**

| Sampling site | Bulk soil (<2 mm)[a] | Clay fraction (<2 μm)[a] |
|---|---|---|
| *Klamath Mountains* | | |
| Eunice Bluff | A, C, G, O, P, Q, S, Ta | A, C, G, P, Q, S, Ta |
| Eunice Bluff parent material | A, C, En, Mg, O, S | |
| Deadfall Lake | A, C, O, P, S, Ta | A, C, G, P, Q, S, Ta |
| Deadfall Lake parent material | A, B, C, P, S, Ta | |
| Swift Creek Late | A, C, P, Q, S, Ta | A, C, G, P, Q, S, Ta |
| Swift Creek Late parent material | O, S | |
| Swift Creek Middle | A, C, P, Q, S, Ta | A, C, G, Q, S, Sm, Ta |
| Swift Creek Middle parent material | O, S | |
| String Bean Creek | Mg, P, Q, S, Sm | G, P, Q, S, Sm |
| String Bean Creek parent material | O, S | |
| *Tablelands* | | |
| Devil's Punchbowl | C, O, Q, S, Ta | C, G, Q, S |
| Winterhouse Gulch Canyon | C, O, Q, S, Ta | G, Q, S |
| Winterhouse Gulch Mouth | C, O, Q, S, Ta | G, Q, S |
| Trout River Gulch | C, O, Q, S, Ta | G, Q, S |
| Tablelands parent material | C, Mg, O, S, Ta | |
| *Pickhandle Gulch* | | |
| Summit | C, Ca, M, P, Q, S, Sm | C, Ca, M, P, Q, S, Sm, Ta |
| Footslope 1 | C, Ca, M, P, Q, S, Sm | C, Ca, M, P, Q, S, Sm, Ta |
| Footslope 2 | C, Ca, M, P, Q, S, Sm | C, Ca, M, P, Q, S, Sm, Ta |
| Pickhandle Gulch Parent Material | C, Mg, S | |

*A* amphibole, *B* biotite, *Ca* calcite, *C* chlorite, *E* enstatite, *G* goethite, *Mg* magnetite, *M* muscovite, *O* olivine, *P* plagioclase, *Q* quartz, *S* serpentine minerals[c], *Sm* smectites[b], *Ta* Talc.
[a]Bulk soil and oriented mount clay-size analysis utilized a Cu Kα instrument. Randomly oriented clay-size analyses utilized a Co Kα instrument. Figures demonstrating phase identification in XRD patterns can be found in supplementary information section S4.
[b]Differentiation between phyllosillicate phases and confirmation of the presence or absence of smectites was confirmed for all soils by oriented mount analysis (Section S4 in supplementary information).
[c]Differentiation of serpentine polymorphs within XRD patterns is given in supplementary information section S4, and individual serpentine minerals based on that identification were used within Rietveld refinements (supplementary information section S5).

Klamath Mountain soils (~12–15 ka). The relatively limited presence of parent material minerals combined with a lack of magnetite in Pickhandle Gulch soils (Table 1, Figs. S22–S24) is consistent with these arid soils primarily incorporating eolian material. Taken together, these trends in the persistence of parent material mineralogy are indicative of climatic differences in aqueous alteration and the release of Fe, Mg, and Si at wetter sites for the formation of secondary materials.

## Relevance of the studied field amorphous material to amorphous material at Gale Crater, Mars

For the studied field samples to be relevant to the CheMin measured samples, and therefore Martian processes, they must exhibit similar in situ formation of Fe- and Si-rich and Al-poor X-ray amorphous material. Fe- and Si-rich and Al-poor amorphous material were observed directly via TEM observations of Klamath Mountain and Tablelands soil clay-size fraction material (Fig. 3). *d*-spacings of ~2.45 and 2.10 Å from nanocrystallites do not match ideal structures for known nanominerals but match at least two XRD peaks consistent with the presence of ferrihydrite[77] and/or

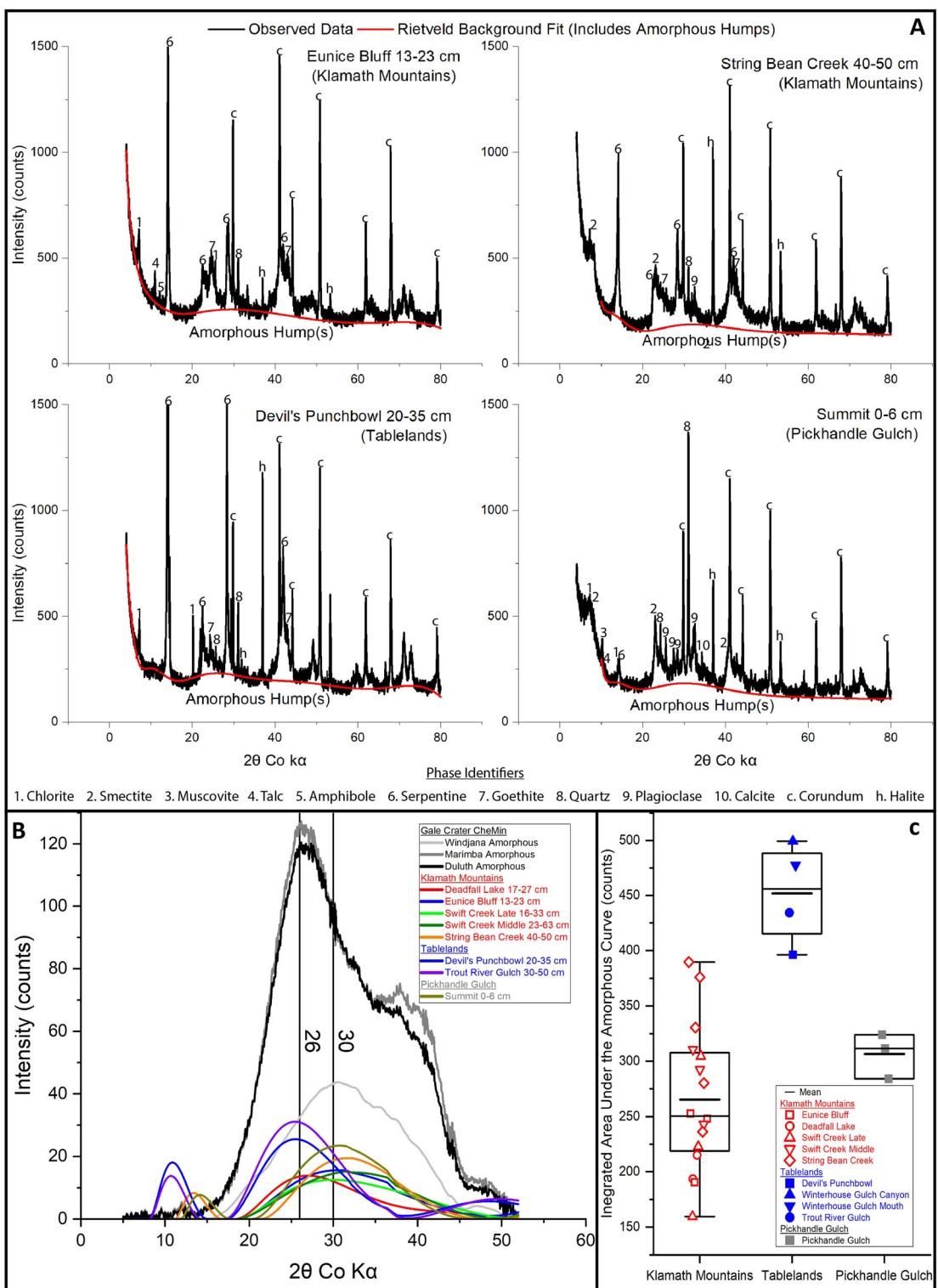

**Fig. 1 | X-ray amorphous contributions to terrestrial soil and Gale crater XRD patterns. A** XRD pattern (Co Kα) showing crystalline phases and the presence of an amorphous hump in the Rietveld fitted background from the clay-size fraction of selected soils; an amorphous hump is observed in the Rietveld fitted background of the clay-size fraction from all soils (Figs. S6, S9, S12, S15, S18, S21, S24). Some peaks attributed to serpentine extend beyond the y-axis scaling in the Eunice Bluff and Devil's Punchbowl soils. **B** Amorphous humps from the clay-size fraction of ultramafic soils compared to Grater crater samples showing the similarity in the amorphous hump position. The selected Gale crater samples possess the most Fe-rich amorphous material observed to date[1–3]. Gale crater amorphous material "humps" are generally centered between 22° and 26°2θ[1–3] with amorphous material richer in Fe and poorer in Si having humps centered at ≥26°2θ[1]. Important caveats should be noted for this panel. The heights and locations of the Rietveld background fits are not Fhkl weighted and should not be taken as directly indicating a particular amorphous material, nor are they directly comparable to the plotted Gale crater amorphous material fits as those plots show directly modeled amorphous material contributions from FULLPAT fitting of Gale crater XRD samples[1,2]. **C** The relative contribution of X-ray amorphous material to all clay-size fraction samples examined by XRD was determined as the integrated area under the amorphous hump showing the highest amount of X-ray amorphous material in the Tablelands soils, consistent with the preferential formation and persistence of this material under cold and wet conditions.

goethite nanoparticles[78]. However, the evidence for amorphous material in the Gale crater comes from broad humps in CheMin XRD patterns[1,4]. Amorphous hump positions are dependent on composition[1,79,80], with increasing $SiO_2$ content correlating with larger *d*-spacings for amorphous silicates[79]. Amorphous humps were present in the backgrounds from Rietveld fits of all clay-size fraction XRD patterns (selected as secondary amorphous material has been observed to concentrate in the clay-size fraction[36]) (e.g., Fig. 1A) and were centered between 26° and 30°2θ (Fig. 1B). While not identifying a specific amorphous material, the terrestrial amorphous humps are centered at similar d-spacings as Gale crater amorphous material[1,2] (Fig. 1B). Relative amorphous material abundances determined from integrating the area underneath the amorphous hump in clay-size fraction XRD patterns indicate greater abundances of amorphous material

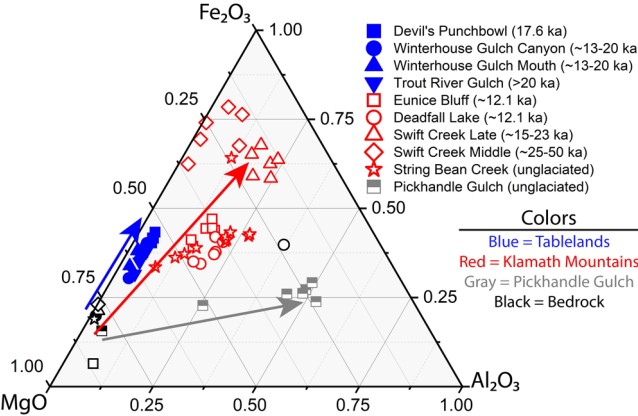

**Fig. 2 | Ternary diagram comparing MgO, $Fe_2O_3$, and $Al_2O_3$ content within the soil clay-size fraction and parent material.** Plotted samples are from all sampled depth intervals within each soil pit in the Klamath Mountains, Tablelands, and Pickhandle Gulch. Total MgO, $Fe_2O_3$, and $Al_2O_3$ content is normalized to 1; numerical values are given in Supplementary Data 2. Arrows denote overall concentration trends from the bedrock to the clay-size fraction material. In the Klamath Mountains and Tablelands, Fe is concentrated into the clay-size fraction relative to parent material while Mg is depleted, reflecting in situ formation of Fe-rich secondary products including amorphous material and Fe-(oxyhydr)oxides coupled with dissolution of Mg-containing primary silicates such as olivine. Some Al-enrichment in the String Bean Creek soil clay-size fraction in the Klamath Mountains likely reflects the formation of smectites in these older, unglaciated soils (Table 1). The Al-content of the Deadfall Lake parent material reflects a more Al-rich plagioclase lherzolite bedrock present at the site which may not reflect the entirety of the parent material of the soil. At Pickhandle Gulch, the major chemical trend involves the concentration of Al in the clay-size fraction relative to parent material concentrations, reflecting the contribution of dust influx of Al-rich phases (e.g., muscovite and smectites) to soil development at that site.

in the subarctic Tablelands (Fig. 1C), consistent with a climatic impact on amorphous material abundance. Bulk dissolution of the clay-size fraction from the Klamath Mountains and Tablelands sites, although not of purely amorphous material, resulted in similarly Al-poor (0.27–11.67 wt% $Al_2O_3$) and Fe- (12.87–23.19 wt% $Fe_2O_3T$) and Si-rich (13.08–50.20 wt% $SiO_2$) chemical compositions as the amorphous materials found in Gale crater, Mars. Increased Fe-content in clay-size fraction material in the wetter Klamath Mountain and Tablelands soils relative to parent material (Fig. 2) is consistent with precipitation of Fe-rich secondary phases, including amorphous material. Amorphous material in these terrestrial soils thus appears to possess similar chemical compositions and amorphous hump shapes and positions as that found at Gale Crater, while exhibiting potential differences in abundance. We therefore examine the conditions that promote formation and persistence of Fe-containing amorphous material, and the implications of such material for Mars.

## The effect of climate and time on the formation and crystallinity of Fe-containing amorphous material

To examine the effect of climate and time on the formation and persistence of Fe-containing amorphous material, the abundance of Fe and Fe-associated Si within amorphous material at each field site was compared using a hydroxylamine hydrochloride dissolution ($Fe_H$ and $Si_H$), which is a combined reduction and acid protonation technique selective for Fe containing amorphous material[74,81–85]. The amount of Fe within total amorphous and crystalline (oxyhydr)oxides was examined using a citrate dithionite dissolution ($Fe_D$), which acts as a reducing agent for $Fe^{3+}$ in $Fe^{3+}$-containing amorphous and crystalline oxides at near-neutral pH conditions[74,86–89]. Together, these measurements allow both a comparison of the amount of Fe in amorphous material and the extent of crystallinity of Fe-containing phases under different climate conditions in soils of different ages. The ratio of ammonium oxalate extractable Fe to dithionite extractable Fe is commonly used as a rough crystallinity index for secondary Fe-containing material[90–100]. Given the similar results produced by hydroxylamine and oxalate extractants[74,84,85] we have substituted the hydroxylamine values for oxalate in the index.

The hydroxylamine hydrochloride and citrate dithionite dissolutions indicate substantially greater quantities of amorphous Fe, as well as greater amounts of amorphous Si (Fig. S45), in the subarctic Tablelands soils than in the warmer Klamath Mountain soils, and a higher degree of Fe-crystallinity within the warmer Mediterranean climate Klamath Mountain soils. Amorphous Fe in the Tablelands soils is ≥1.8× parent material values (Fig. 4), indicating in situ formation of Fe-containing amorphous material. Similarly, the ratio of amorphous Fe to the Fe in oxyhydroxides (both amorphous and crystalline) in the Tablelands is > 0.95 (Fig. 4), suggesting that almost all secondary Fe-containing material in the Tablelands soils is poorly crystalline to amorphous. In contrast, amorphous Fe within the ~12–50 ka Klamath Mountains soils overlapping in age with the Tablelands

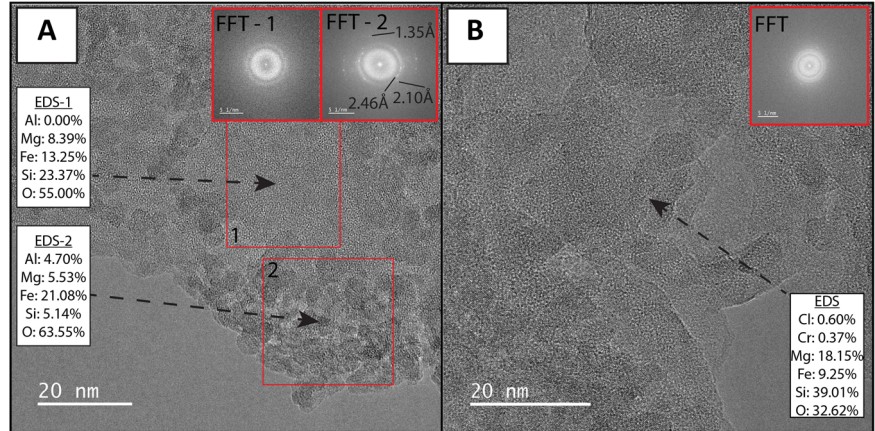

**Fig. 3 | TEM imagery of X-ray amorphous material.** Amorphous and nanocrystalline materials were confirmed through fast Fourier transform (FFT) scans presented as inset boxes. **A** Intermixed truly amorphous and nanocrystalline materials from the clay-size fraction from the BC horizon (23–33 cm depth interval) in the Eunice Bluff soil in the Klamath Mountains. Zones of nanocrystalline packets (FFT-2, EDS-2) contain elevated Fe content relative to truly amorphous gel material (FFT-1, EDS-1). **B** Amorphous gel material from the clay-size fraction from the Devil's Punchbowl soil (C Horizon, 20–35 cm depth interval).

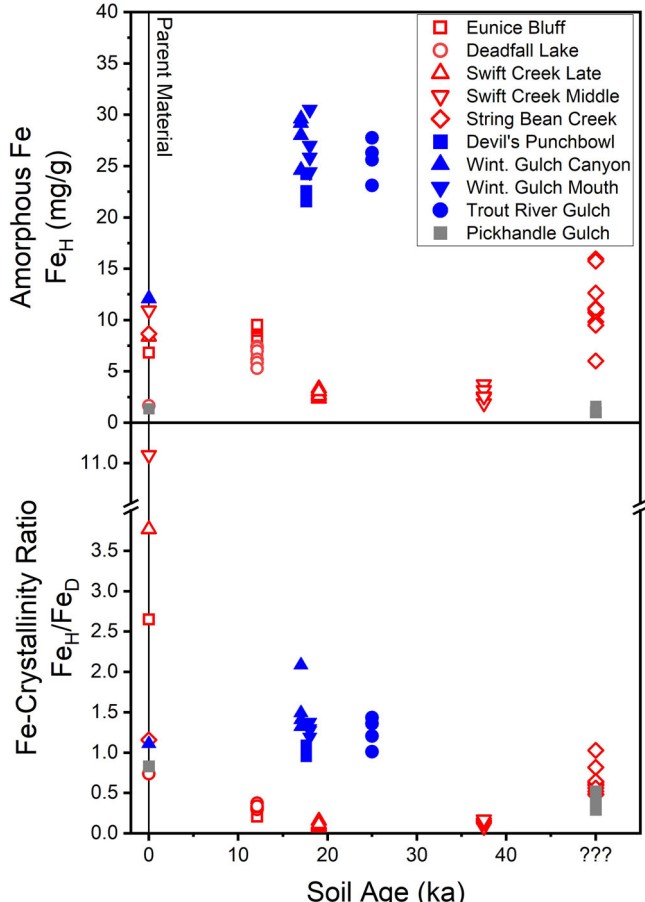

**Fig. 4 | Selective dissolution results for Fe from bulk soil material plotted vs soil age.** Color denotes sampling location; Klamath Mountain samples are in red, Tablelands in Blue, and Pickhandle Gulch in gray. The Tableland soils possess total amorphous Fe ($Fe_H$ in mg/g soil) than the Klamath Mountain soils and exhibit an increase in total amorphous Fe relative to parent material values, while in the Klamath Mountains and at Pickhandle Gulch, the amorphous Fe is the same as or less than parent material amorphous Fe. Fe within the Tablelands soils is also primarily found within amorphous material ($Fe_H/Fe_D > 0.95$), whereas Fe within the Klamath Mountain and Pickhandle Gulch soils is primarily found within crystalline secondary phases. Data points represent all sampled depth intervals within each soil profile (see Supplementary Data 3). Parent material values are given at time = 0 ka. Klamath Mountain sampling sites are geographically separated (Fig. S1), and one parent material sample was analyzed per soil pit. In contrast, soil pits in the Tablelands and Pickhandle Gulch regions are geographically close to one another (Fig. S1). One parent material sample was analyzed each for the Tablelands and Pickhandle Gulch regions.

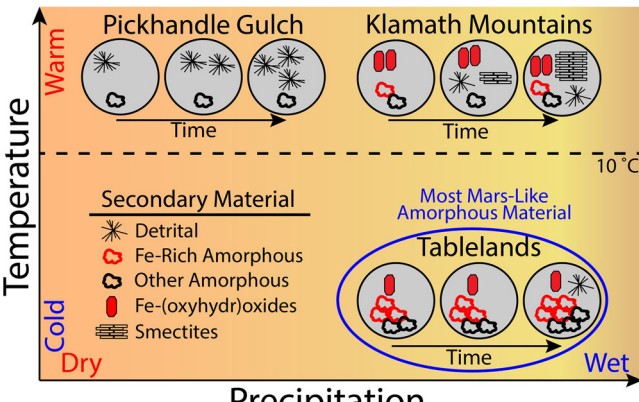

**Fig. 5 | Conceptual model of the impact of climate upon the development of secondary material within the examined Fe-rich soils.** In the hot and arid climate at Pickhandle Gulch, the only observed accumulations of non-parent material at this site can be attributed to detrital input of dust-borne particles. In the relatively warm and wet Mediterranean climate of the Klamath Mountains, the formation of Fe-oxyhydroxides occurs rapidly (within ~12 ka), with the development of smectites observed in older soils (25–50+ ka), with relatively limited in situ formation and persistence of amorphous material. In the subarctic climate of the Tablelands, amorphous material (both Fe- and Si-containing) forms rapidly and is the predominant secondary material in all examined soils. Smectite development was not observed in any measured soil in the Tablelands.

be consistent with the presence of nanocrystalline or poorly crystalline smectite precursor phases[51]. However, even in this Klamath Mountain soil richest in amorphous Fe, mean amorphous Fe (11.34 mg Fe/g soil) is still less than half that in the Tablelands soils (25.77 mg Fe/g soil), despite similar parent material values (Tablelands = 12.11 mg Fe/g soil, String Bean Creek = 8.68 mg Fe/g soil). Substantially more abundant Fe-containing amorphous material and lower Fe-crystallinity in Tablelands compared to Klamath Mountain soils thus remain consistent with the subarctic climate promoting greater formation and longer persistence of Fe-containing amorphous material.

Amorphous and oxyhydroxide-contained Fe-values from the Pickhandle Gulch soils are roughly equivalent to parent material values, and the low ratios of amorphous Fe to Fe in oxyhydroxides (Fig. 4) together indicate high secondary Fe-crystallinity and minimal to no in situ formation of Fe-containing amorphous material. The dry desert climate at Pickhandle Gulch, therefore, seems to preclude much in situ formation of Fe-containing amorphous material.

### Implications for past Martian conditions

The amorphous and crystalline secondary weathering products in these Fe-rich and Al-poor soils developing under a range of climate conditions (Mediterranean, subarctic, and desert), and over a range of ages (~11 to >50 ka) can help interpret the climatic conditions consistent with the widespread presence of Fe-rich amorphous material at Gale crater and elsewhere on Mars (Fig. 5). The minimal Fe-containing secondary material in soils at the desert climate Pickhandle Gulch site suggests wet conditions are necessary for in situ formation of Fe-containing secondary material. Among the wetter field sites, the cooler conditions at the Tablelands favor the formation and persistence of amorphous material, particularly Fe-containing amorphous material, whereas the warmer conditions in the Klamath Mountains favor the development of crystalline phases (Fig. 5), Fe-(oxyhydr)oxides in the youngest soils and smectites as soil age increases.

While there is a range of possible formation mechanisms for the amorphous material at Gale crater[4] and chemical variability potentially suggests multiple source materials[5] and alteration conditions[4,101], we demonstrate that cooler conditions such as those at the Tablelands preferentially form and preserve Fe-rich amorphous material. The production

soils is generally equal to or less than that found within parent material, and the ratio of amorphous Fe to the Fe in oxyhydroxides is lower than that ratio in parent material (Fig. 4), consistent with minimal formation of Fe-containing amorphous material or conversion of amorphous material into crystalline minerals. Fe-containing secondary material is, therefore, almost entirely amorphous in the subarctic Tablelands soils, while the warmer conditions of the Klamath Mountains promote crystalline Fe-containing secondary material formation.

In the likely older (undated but unglaciated) String Bean Creek Klamath Mountain soil, higher amorphous Fe ranges (6.02–15.96 mg Fe/g soil) and ratios of amorphous Fe to Fe in oxyhydroxides (Fig. 4) indicate both greater amounts of amorphous Fe-containing material and overall lower Fe-crystallinity compared to the younger Klamath Mountain soils. This increase in amorphous Fe-containing material coupled with abundant smectites detected by XRD not found in the youngest soils (Table 1) might

of abundant Fe-containing amorphous material at Gale crater[1] may, therefore, also be consistent with wet and near-freezing mean annual conditions. The abundant Fe-rich amorphous material within Gale Crater thus bolsters arguments[35,102–105] for the presence of wet but cold conditions during formation followed by long-term cold and dry conditions facilitating preservation. The presence of similar nanophase Fe-oxides[18,19] and amorphous Fe-silicates elsewhere on Mars[13] would likewise be consistent with cool and wet conditions during formation under a generally cold and icy climate. Returned samples from Mars will allow a detailed comparison of Martian amorphous material with material such as that examined here to better understand past Martian environments.

## Methods
### Field sites
The Trinity Ultramafic Body covers ~2100 km$^2$ in the eastern Klamath Mountains and is composed of a partially serpentinized mélange of harzburgite and lherzolite (60–70%), dunite (15–20%), and plagioclase lherzolite (10–15%) units cut by clinopyroxene-rich dikes in some areas[106,107]. Formation of the peridotite sections of the Trinity Ultramafic Body likely occurred around 472 ± 32 Ma, with emplacement of the body occurring during the early Devonian[108,109]. The Rattlesnake Creek Terrane lies southwest of the Trinity Ultramafic Body and is composed of a mixed sequence of serpentinite basement unconformably overlain by Upper Triassic and Lower Jurassic volcanic, hemipelagic, and clastic units with intruded gabbroic and quartz diorites throughout that trend into amphibolite facies and locally higher metamorphic grade materials[108,110,111]. The serpentinized basement consists of peridotite, greenstone, amphibolite, and pillow basalts that have undergone varying degrees of serpentinization[110]. The genesis of the Rattlesnake Creek terrane ophiolite and associated facies and their emplacement timing remain poorly understood[108].

Samples were collected from five sites within the Klamath Mountains, four sites within the Trinity Ultramafic Body (two sites in high-altitude cirques near Mt. Eddy dubbed Eunice Bluff and Deadfall Lake and two sites within the Swift Creek Valley), and one site within the Rattlesnake Creek Terrane dubbed String Bean Creek (Fig. S1). The Trinity Alps, a formerly glaciated region of the Northeast Klamath Mountains where several formerly glaciated valleys feature serpentinite bedrock[112], were most recently extensively glaciated coterminous with the late Wisconsinan glacial period of the Laurentide Ice Sheet[112]. Recent work using cosmogenic techniques has confirmed Younger Dryas deglaciation in high-altitude cirques within the Trinity Alps by ~12.1 ka[113,114]. The deglaciation date of ~12.1 ka is used as the initiation date for pedogenesis in the high-altitude cirque soils at the Deadfall Lake and Eunice Bluff sites, while within the Swift Creek valley, samples were collected from sites determined by Sharp (1960) to be of Late and Middle Wisconsinan ages. The Late Wisconsinan Cordilleran Ice sheet reached its maximum extent sometime between about 15–23 ka, with initiation of retreat around ~15.6 ka[115], with Middle Wisconsinan dates ranging roughly between 25-50 ka[116,117]. As moraines within Swift Creek lack cosmogenic dates, we use these age ranges as approximate estimates for the ages of the Late and Middle Wisconsinan features described by Sharp (1960)[112]. In contrast to the Trinity Ultramafic Body, previous glaciation has not been established within the Rattlesnake Creek Terrane, and the soil sampled there is undated but likely substantially older than the soils in the Trinity Ultramafic Body.

The Gros Morne Tablelands is one of four highland plateaus comprising the Bay of Islands ophiolite, part of the Humber Arm Allochthon located on the western flank of the Island of Newfoundland, Canada[118–120]. The Bay of Islands ophiolite was emplaced onto the passive continental margin of eastern North America during the Ordovician age Taconian Orogeny[119,121]. The plateau edges are defined by sharp u-shaped formerly glaciated valleys with associated glacial landforms, including moraines, rock glacier deposits, pro-talus lobes, and diamict[118,122]. Vegetation on the Tablelands is sparse, composed primarily of small ground-hugging shrubs with stunted pine trees at lower elevations.

Soil samples were collected from three formerly glaciated valleys on the Tablelands (Fig. S1). One sampling site was located just up-slope from and within the protruding lobe of a terminal moraine within the Devil's Punchbowl cirque. The terminal moraine was dated to 17.6 ± 0.3 ka using chlorine-36 isotope analyses[122]. One sampling site was located on a terrace overlooking Highway 471 in Trout River Gulch. Debris benches in the Trout River Gulch valley were cosmogenically dated using $^{36}Cl$ isotopes to between 15 and 20 ka[122], necessitating the establishment of an ice-free valley prior to that point. Quartzite erratics sourced from over 20 km east of Bonne Bay necessitate westward ice flow at the time of deposition[118]. As Late Wisconsinan ice flow in the area was directed northward[118,123], this westward flow through Trout River Gulch ceased prior to the most recent glacial maximum. The combination of $^{36}Cl$ ages and glacial erratics indicate that Trout River Gulch was likely not glaciated during the Last Glacial Maximum[118,122]. Finally, two sampling spots were located on diamict deposits in Winterhouse Gulch, a previously glaciated tributary valley that feeds into the Trout River Gulch. One sampling site was located at the mouth of Winterhouse Gulch, and one halfway up the valley between the mouth and the cirque. Though cosmogenic ages have not been determined for glacial material in Winterhouse Gulch Canyon, the late Wisconsinan age ranges for other glacial deposits on the Tablelands, suggest a similar age range is likely for the presence of a valley glacier in Winterhouse Gulch[122], and thus for initiation of soil development following glacial retreat. Reconstructions of Late Wisconsinan ice in Atlantic Canada suggest that the icesheet covering Newfoundland reached a most recent maximum extent around 20 ka[123,124], followed by deglaciation that resulted in glaciers retreating away from the coast by ~12 ka[123,124]. Initiation of soil development in Winterhouse Gulch likely falls within this age range, concurrent with all other cosmogenically dated glacial landforms surrounding the Tablelands[122], with the soil at the mouth of Winterhouse Gulch likely older than the soil closer to the cirque from retreat of the glacier that formed the valley.

Pickhandle Gulch is a ghost town located in the now-defunct Candelaria mining district roughly 50 km northwest of Tonopah, Nevada. A serpentinite body roughly 6800 ft by 300–1350 ft is located southeast of the Northern Belle and Mount Diablo open pit mines with smaller serpentinite lenses scattered nearby, with some exposures of serpentinite within the nearby Northern Belle mine workings[125]. Soil cover overlying the serpentinite bedrock is thin (<10 cm) and is mantled by a desert pavement composed of serpentinite clasts, with a sparse vegetation cover primarily composed of sagebrush. Previous literature examining the serpentinite body at Pickhandle Gulch is extremely limited, and to our knowledge, only one report from the Nevada Bureau of Mines attests to the serpentinite's presence[125]. The serpentinite is bounded by faults, with a larger mass likely to exist unexposed in the subsurface[125]. The age and provenance of the serpentinite are currently unknown; however, the body contains inclusions of lower Triassic Candelaria shale material[125]. The exposure age of the serpentinite is unknown. At Pickhandle Gulch, the serpentinite body is bounded by sharp contacts with fluvial channels at its base, and toeslope locations could not be identified. Soil samples were therefore collected from a hill summit position and from two footslope locations to provide a degree of spatial variation as time differentiation between soils likely does not exist in a similar manner as in the Klamath Mountains or Tablelands (Fig. S1).

### Sample collection
Within the Klamath Mountains and Tablelands, soil pits were located on toeslopes at the base of large hillsides as these have previously been shown to accumulate amorphous material and secondary clay minerals[51], and soil development along a hillside is typically greatest at footslope/toeslope positions[126]. In the Klamath Mountains, there is extensive tree cover, and we chose to locate pits not immediately adjacent to trees to limit the need to cut through large root structures and avoid such overt biological impacts. In each location, soil pits were excavated by hand using picks and shovels to within a C horizon, contact with the bedrock, point of refusal, or contact with standing water. In the Klamath Mountains, the Eunice Bluff and Deadfall Lake soil pits were excavated to the point of refusal, while the Swift

Creek Late, Swift Creek Middle, and String Bean Creek soils were excavated until contact with a C horizon determined by decreasing soil redness in relation to overlying soil material[127]. In the Tablelands, soil pits were excavated until infilling water was observed in the Winterhouse Gulch Canyon, Winterhouse Gulch Mouth, and Trout River Gulch sites and to contact with bedrock at the Devil's Punchbowl site. As a result of the shallow nature of the Pickhandle Gulch soils (<~10 cm), soil pits in that field location were excavated into contact with the bedrock.

We collected at least one soil sample from within each soil horizon (see Supplementary Data 3). In the Eunice Bluff, Deadfall Lake, and String Bean Creek soils, Munsell Color variation allowed for the identification of a clear organically enriched A horizon at the surface of each soil pit. We elected to sample in 10 cm depth intervals below the A horizon at those three sites to increase the density of sampling. At the Swift Creek sites in the Klamath Mountains (Late and Middle), one sample was collected from each soil horizon identified in the field. In the Tablelands soil pits, relatively uniform Munsell soil color indicated that A and B horizons had not formed. Because the Tablelands soil profiles were visually undifferentiated, we gathered soil material from four depth intervals within each Tablelands soil pit to provide a higher density of sample material and evaluate possible variations in soil properties with depths that were not visually observable. At Pickhandle Gulch, the presence of small pores indicated the development of Av vesicular surface horizons. We sampled from the Av horizons and from thin underlying C horizons that lacked similar porosity development. When referring to our samples in the methods, main text, and supplementary information, we sometimes refer to sampled depth intervals and not to soil horizons, as we sampled multiple different intervals from within some soil horizons in three of the Klamath Mountain sites and from all four Tablelands soils (see Supplementary Data 3).

A minimum of ~4 kg of combined bulk soil and gravel material was collected from within each sampled depth interval at all soil pit sampling sites. Parent material bedrock samples were collected from buried clasts from within each soil pit. The collected bulk soil and gravel samples from the Klamath Mountains and Pickhandle Gulch were stored on dry ice during transport and immediately placed in a −20 °C freezer upon return to UNLV. Bulk soil and gravel samples from the Tablelands were packaged and flown back to UNLV and placed in a −20 °C freezer upon arrival but were not packaged on dry ice during transport.

## Parent material sample preparation

We processed one parent material sample each from the Tablelands and Pickhandle Gulch locations as soil pits at these sites were geographically close and developing out of the same ultramafic body. We processed a parent material sample from each Klamath Mountain sampling site as these sites were geographically disparate and developing out of varied ultramafic material. Each parent material sample was trimmed to remove weathering rinds using a cutting saw, crushed within a Bico Chipmunk Jaw Crusher Model 241-36 WD to produce chunks about 3 mm in diameter, and then powdered in a Fritsch pulverisette model 02-101 agate mortar grinder mill. The saw blade used to trim samples was cleaned between uses using a 100% ACS reagent grade ethanol wash. During crushing in the chipmunk jaw crusher, and then again when powdering in the pulverisette, two sub-samples of material were run to pre-contaminate the equipment, and those sub-samples were then discarded. A third crushing or powdering run was then conducted using the pre-contaminated instruments and then collected for X-ray diffraction and bulk and selective chemical dissolution analyses. The jaw crusher was thoroughly cleaned between samples using a compressed air blower and a 100% ACS reagent grade ethanol wash. The pulverisette was cleaned between samples with a 100% ACS reagent-grade ethanol wash. Following powdering, parent material samples were stored in a −20 °C freezer.

## Soil sample preparation

After removal from the freezer, each combined soil and gravel sample was first air-dried, and the < 2 mm bulk soil fraction was sieved from the overall sampled material. Gravel content was estimated by massing the

<2 and >2 mm fractions following sieving. Sub-samples of bulk soil material were then powdered in a Fritsch pulverisette model 02-101 agate mortar grinder mill prior to XRD analysis and application of selective chemical dissolutions. Two sub-samples of bulk soil material were powdered first to pre-contaminate the pulverisette and discarded afterward, and a third sub-sample was then powdered and collected for XRD, bulk chemical dissolution, and selective chemical dissolution analyses. The pulverisette was thoroughly washed between samples using 100% ACS reagent grade ethanol. Both powdered and unpowdered bulk soil samples were stored at −20 °C following powdering.

The clay-size fraction (<2 μm) was separated from the bulk soil following a modification of the methods in Edwards and Bremner (1967)[128]. Briefly, ~10 g of soil was mixed with 25 mL of 18.2 MΩ DI water, sonicated for 30 min, vortexed, and allowed to settle for 1.5 h. The suspended load was then pipetted off, NaCl (JT Baker ACS Reagent Grade) was added to the supernatant to a concentration of 1 M to promote flocculation, and the samples were allowed to settle for ~30 min. Following this, the samples were centrifuged for 3 min at 8000 rpm (8228×g), and the clear supernatant was decanted from the clay-size fraction pellet. The clay-size fraction pellets then were washed with 18.2 MΩ DI water and centrifuged 3 more times to remove residual salt, frozen at −20 °C for at least 12 h, and then freeze-dried (LabConco Freezone 4.5 Freeze Dry System).

## Mineralogical analyses and quantification

XRD analyses of randomly oriented powdered bulk soil samples were performed at the UNLV Geosciences Shockwave laboratory on a Proto-AXD Bragg-Brentano type X-ray diffractometer with a Cu K-alpha tube (1.541 Å wavelength), dectris hybrid pixel array detector, 0.4 mm soller slits, and a Ni filter between 4° and 70°2θ. A knife-blade attachment was added to limit the over-saturation of the detector at low angles. Randomly oriented mounts were prepared by top loading the sample into brass or zero background (MTI corp. ZeroSi24D10C1cavity10D) sample stages. Excess material was gently scraped off the top using a razor blade to limit the preferred orientation of phyllosilicates. Mineral identification within randomly oriented bulk soil XRD patterns was determined using the program X'Pert HighScore[129] and through comparison with d-spacings of crystal lattice planes reported in the American Mineralogist RRUFF database[130] and the crystallography open database[131].

Randomly oriented clay-size fraction samples extracted as described above were spiked to contain 20 wt% 0.8–1-μm diameter α-Al2O3 (Beantown Chemical 127075-25G) and analyzed from 3° to 80°2θ at the X-ray Diffraction Laboratory at NASA Johnson Space Center in Houston, Texas on a Malvern PANalytical X'Pert Pro MPD X-ray Diffractometer equipped with a Co K-alpha source (45 kV, 40 mA, ½° anti-scatter slit, ¼° fixed divergence slit, 0.01067° step size, and 0.5 s/step). Spiked samples were prepared by combining pre-weighed α-Al$_2$O$_3$ and powdered parent material or clay-size fraction material and mixing the two materials together within an agate mortar and pestle by a combination of pressing material toward the center and a circular motion. Mineral identification within randomly oriented clay-size fraction XRD patterns was determined by using the programs X'Pert HighScore[129] and Profex[132] and through comparison with d-spacings of crystal lattice planes reported in the American Mineralogist RRUFF database[130] and the crystallography open database[131].

Oriented mount XRD measurements were conducted on a Proto-AXD Bragg-Brentano type X-ray diffractometer in the UNLV Geosciences Shockwave laboratory with a Cu K-alpha tube (1.541 Å wavelength), dectris hybrid pixel array detector, 0.4 mm soller slits, a Ni filter, and a knife-blade attachment between 2° and 14°2θ. XRD analyses were conducted on oriented clay-size fraction material following cation saturation and various chemical and heat treatments to determine the presence or absence of expandable 2:1 clay minerals in each soil. Phyllosilicate mineralogy determinations from oriented mount analysis, as described below, assist with determinations of phyllosilicate mineral presence, and particularly the presence of expandable 2:1 clay minerals, within XRD patterns acquired from randomly oriented mounts.

To identify whether expandable 2:1 clay minerals were present in the clay-size fraction, oriented mounts of the clay-size fraction were analyzed by XRD after 3 different treatments: (1) Saturation with $Mg^{2+}$ followed by air-drying, (2) ethylene glycol vapor solvation of the air-dried $Mg^{2+}$ saturated mounts, and (3) $K^+$ saturation followed by heating to 550 °C (see Supplementary Data 3). $Mg^{2+}$ or $K^+$ saturations were prepared[47,133–136] by suspending 300 mg of clay-size fraction material in solutions of either 1 N $MgCl_2$ (VWR ACS Reagent grade CAS 7791-18-6) or 0.1 N KCl (OmniPur EMG Chemical Argentometric grade CAS 7447-40-7) in 18.2 MΩ water, shaking intermittently by hand for ~1 h followed by centrifugation for 3 min at 8000 rpm (8228×*g*), after which the supernatant was decanted, and the samples then washed with a 50:50 18.2 MΩ DI water-to-ethanol mixture and centrifuged for 3 min at 8000 rpm 3 times to remove excess salts.

Three mL of 70% ethanol 30% water solution (Aldon Corp. EE0069 Denatured Ethyl Alcohol) was then added to each sample, and the container vortexed to suspend the clay-size particles. ~1 mL of the resulting slurry was then pipetted onto fused silica slides (Alfa Aesar Quartz microscope slides, Part #42295) and allowed to air dry in a covered container overnight in a fume hood.

Previous methods have called for 3–5 overnight wash cycles using cation-saturated solutions followed by rinsing[134,137], or direct application of cation solution to already oriented clays through a suction device onto ceramic slides followed by a DI-water rinse[136]. To test whether a 1-h treatment produced similar saturation to the 3+ day treatment, we compared XRD patterns from a 3-day and 1-h Mg-saturation procedure. We observed similar peak locations between air-dried and glycolated XRD patterns from the two saturation durations indicating that a 1-h saturation procedure produces cation saturation like a 72-hour saturation procedure (Fig. S2).

Mg-saturated mounts were analyzed after air drying and again after solvation with ethylene glycol (VWR ethylene glycol ≥ 99%, CAS: 107-21-1) vapor in a closed container maintained at 60–70 °C for at least 12 h[134,138]. *K*-saturated oriented mounts were analyzed after heating to 550 °C for at least 30 min[138]. Table S2 gives reference d-spacings used to identify the presence of various phyllosilicates within the oriented mount XRD patterns. Smectite (001) planes expand upon solvation with ethylene glycol[134] with the degree of expansion largely controlled by layer charge[133–135,139,140], with typically observed *d*(001) expansion of low-charge smectites following solvation with ethylene glycol to between ~16 and ~18 Å[134,135,141]. High charge smectites and vermiculite typically do not exhibit (001) expansion or exhibit extremely limited expansion to between ~14 and ~14.5 Å following ethylene glycol vapor solvation[133–135,142]. Both high- and low-charge smectite (001) planes collapse to ~10 Å following heating to 550 °C[47,133,134].

Serpentine mineral (lizardite, antigorite, chrysotile) XRD peaks commonly overlap, and distinguishing whether one or multiple serpentine polymorphs are present in powder XRD patterns can be difficult. Lizardite, antigorite, and chrysotile presence in XRD patterns were interpreted primarily through the methodology of Peacock (1987)[107]. Identification of antigorite was based on the lack of asymmetric (020) peaks at ~4.62 to 4.57 Å and the presence of a (16 0 1) peak at ~2.53 Å. Although transmission electron microscopy (TEM) analyses detected chrysotile fibers in the clay-size fraction of four soil samples (Fig. S49), chrysotile identification is difficult and generally not definitive by XRD when multiple serpentine polymorphs are present and in various states of disorder or incorporate variable amounts of Mg, Fe, Al, Ni, or other elements. Determination from XRD alone may not be definitive when mixtures of these phases are present within a sample[107]. Additional complications in distinguishing individual serpentine phases arise from the common co-occurrence of the serpentine polymorphs[143–145] and from overlap of chlorite and talc peaks in the range of the lizardite and chrysotile (020) peaks and from olivine, which has a major (131) peak at ~2.51 Å[107], and from a major goethite peak (111) at ~2.44 Å[146]. While TEM imagery indicates chrysotile in the four soil samples analyzed by TEM, its presence is not definitive based on XRD alone.

Clay-size fraction material from at least one depth interval within A and B (when present) and C soil horizons from all soil pits was analyzed through Rietveld refinement of randomly oriented XRD patterns[147,148] using

Profex 5.1[132], based on the BGMN refinement architecture[149], with structure files from the Crystallography Open Database[131] (see Table S3 in supplementary information for crystal structure files used during refinement). Rietveld refinements were conducted on the clay-size fraction samples spiked with 20 wt% 0.8–1-μm diameter α-$Al_2O_3$ (Beantown Chemical 127075-25G). The Crystallography Open Database crystallographic information files used for Rietveld refinement, as well as the origin of files used for smectites, are given in Table S3. For the Rietveld refinements, XRD patterns were analyzed between 5° and 80°2*θ* for samples that lacked smectite content, or when samples possessed smectites from 10° to 80°2*θ* to exclude the highly variable smectite (001) planes as in Smith et al.[8]. Though chrysotile was present in TEM imagery (Fig. S49), it was not included in Rietveld refinement due to unavoidable significant overlap between major peaks between different serpentine polymorphs. During fitting, we refined unit cell parameters, scale factor, crystallite size, microstrain, and preferred orientation. Halite resulting from the Na-saturation utilized during the clay-size fraction extraction procedure was observed in several clay-size samples utilized for Rietveld refinement. XRD patterns that exhibited the presence of peaks attributable to halite were cut between 31.6° to 32.2°2*θ*, 36.4° to 37.4°2*θ*, 52.6° to 54.0°2*θ*, and 66.2° to 67.2°2*θ* as these represent peaks attributed to halite from comparison with a halite crystallographic information file from the Crystallography Open Database (COD 9006374)[150].

Rietveld refinements were utilized to fit a background to each clay-size fraction XRD pattern for the purpose of determining relative amorphous material contributions to each XRD pattern. Rietveld refinement was not utilized to directly calculate X-ray amorphous material abundances via the internal standard method as we could not accurately estimate the density of the various potential amorphous materials present. The prevalence of phyllosilicate phases with highly overlapping peaks and a high potential degree of disorder and chemical variation in octahedral and tetrahedral sites similarly would induce a high degree of uncertainty in any calculated phase abundances. Instead, relative amorphous contributions were determined by calculating a potential amorphous contribution to the fitted Rietveld background, as amorphous material is typically part of the background fit of XRD Rietveld refinements[7,8]. Following the fitting of a background during Rietveld refinement, a baseline was calculated for each fitted background utilizing an SNIP function with 100 iterations in the Profex program[132]. This baseline was then subtracted from the fitted Rietveld background to produce an amorphous hump pattern. The area under the central peak of each amorphous hump was then integrated into OriginPro software. As each XRD sample was analyzed using identical instrument parameters and processing methods, the relative area underneath each amorphous hump contribution to the fitted Rietveld backgrounds provides a relative indication of the abundance of amorphous material within each clay-size fraction.

To ensure the presence of X-ray amorphous material in the samples, we analyzed select samples of clay-size fraction material from the Klamath Mountains and the Tablelands via high-resolution transmission electron microscopy (HRTEM). We analyzed material from the Eunice Bluff soil BC horizon (23–33 cm depth interval) from the Klamath Mountains and the Devil's Punchbowl soil C horizon (20–35 cm depth interval) from the Tablelands. HRTEM sample preparation was performed as follows. A small amount (~15 mg) of clay-size fraction material was added to ~15 mL of 70% ethanol solution. The solution was then sonicated for 5 minutes to disperse aggregates of clay-size particles. A carbon-coated copper TEM grid was then swirled in the mixture several times and allowed to air dry. Prepared samples were then analyzed using a Titan 300/80 (FEI) aberration-corrected TEM at 300 kV accelerating voltage at Arizona State University. Electron-dispersive spectroscopy results were normalized without Cu or C due to contributions to those elements from the grid material. During our analytical run, the selected area diffraction (SAED) detector exhibited problems and could not be utilized. The presence or absence of amorphous material was determined using fast Fourier transform (FFT) imagery. FFT images produce similar 2D patterns to SAED imagery and can be analyzed in a similar manner. Amorphous material was identified based on the absence of lattice fringes in FFT diffractograms and HRTEM images. Nanocrystalline material, still

falling under the X-ray amorphous component of XRD data[9,10], was identified based on the presence of crystalline packets less than 100 nm across in HRTEM images with diffuse spots in FFT diffractograms.

## Chemical analyses

Bulk soil pH was measured via a 1:1 mass-to-volume ratio of unpowdered <2 mm bulk soil fraction material to 18.2 MΩ water with a Mettler Toledo InLab Expert Pro pH probe[151]. All samples were dried at 105 °C overnight for at least 12 h to dehydrate the samples using ceramic crucibles that had also previously been dried at 105 °C overnight for at least 12 h. Loss on ignition at 550 °C (LOI$_{550}$), a proxy for organic carbon content, was applied to unpowdered bulk material following the method of Heiri et al. [72]. Estimated error for the method is ±2 wt%[72]. Briefly, 5 g of unpowdered air-dried bulk soil material was loaded into oven-dried ceramic crucibles, heated to 105 °C overnight for at least 12 h to dehydrate the samples, cooled and weighed, heated to 550 °C for 4 h to burn off organic carbon, then cooled and weighed again. Loss on ignition at 950 °C (LOI$_{950}$), as in Jones et al. [152], was applied to all clay-size fraction and powdered parent material samples to examine total volatile content. 1 gram of freeze-dried clay-size fraction material or powdered parent material oven-dried at 105 °C overnight for at least 12 h to ensure dehydration, followed by heating to 950 °C for 2 h, allowed to cool overnight, then weighed.

To examine trends in crystallinity of Fe-containing materials within each soil site as well as the bulk chemical composition of the parent bedrock and clay-size fraction material, a suite of total and selective chemical dissolutions was applied to material from all sampled depth intervals in all soil pits (depth intervals and soil horizon designations are given in Supplementary Data 3).

ICP-MS analyses for bulk major and trace element chemistry of the total digestions of the powdered parent material and clay-size fraction samples were measured at the Nevada Plasma Facility Lab (NPFL) at UNLV using a ThermoFisher Scientific™ iCAP Qc ICP–MS after the following digestion—50 mg of sample material was dissolved in acid cleaned Savillex beakers in a 1:1 mixture of sub-boiled double distilled HNO$_3$ and Optima grade HF. Samples were analyzed concurrently with the USGS standard BHVO-1, AGV-1, and BCR-1, and total procedural blanks. Reproducibility and detailed methods of NPFL data are presented in DeFelice et al. [153]. Off-gassing of SiF$_4$ during dissolution of silicates in HF precludes accurate direct measurements of SiO$_2$ in our samples[154]. SiO$_2$ content was instead calculated by subtracting the total major element oxides measured by ICP-MS from 100% as in Udry et al. [155] after incorporating the volatile content determined by loss on ignition at 950 °C from 100% as in Jones et al. [152].

Na$_2$O was excluded from ICP–MS measurements of the clay-size fraction chemistry as the clay-size fraction extraction procedure involved the addition of NaCl to promote flocculation. Na$_2$O content within parent material samples was measured (see Supplementary Data 2) to estimate whether exclusion of Na$_2$O would substantially alter calculated SiO$_2$ values, and minimal Na$_2$O content is present in the parent material samples for Pickhandle Gulch (0.00 wt%), the Tablelands (0.02 wt%), and for the Eunice Bluff (0.06 wt%), Swift Creek (0.01 wt%), and String Bean Creek (0.02 wt%) sampling sites in the Klamath Mountains. Na$_2$O content was slightly higher within the Deadfall Lake parent material (1.40 wt%) because of the presence of plagioclase, likely including albite, not observed in the bedrock at other sites and likely not contributing to this extent to the soil formed at Deadfall Lake based on the decreased Al present in the clay-size fraction from Deadfall lake relative to the parent material, and the presence of olivine in the clay size fraction despite the absence of olivine in the parent material (Fig. S3 and Table 1). As SiO$_2$ content is calculated based on the measured major element oxide concentrations, not including Na$_2$O, it should be read as a potential maximum value and is likely very accurate for the soil pits with minimal parent material Na$_2$O values, but potentially slightly over-estimates the SiO$_2$ content for the Deadfall Lake soil. Additionally, if plagioclase found within some soils included a significant albite component, this would also lead to overestimates of the SiO$_2$ content. As Na is commonly incorporated

into smectites[134] the presence of smectites could also potentially contribute to an overestimate of SiO$_2$ content.

Selective chemical dissolution techniques were applied to powdered bulk soil samples, oven dried at 105 °C for 12 h to ensure dehydration, from all sampled depth intervals from all soil pits (see Supplementary Data 3) to examine the distribution of secondary Fe within different soil reservoirs. Hydroxylamine hydrochloride, citrate dithionite, and sodium pyrophosphate extractants were applied to study Fe within amorphous material[74,83,84], amorphous and crystalline (oxyhydr)oxides[74,89,151], and organically bound[73,74] Fe-reservoirs respectively. The hydroxylamine hydrochloride supernatant was also analyzed for Si content.

A sodium pyrophosphate extraction was applied to analyze organically complexed Fe (Fe$_P$) following the method delineated within Shang and Zelazny (2008)[61] a modification of the method used by McKeague (1967)[63]. Whether the pyrophosphate technique results in the dissolution of purely organic compounds or whether the supernatant potentially includes some Fe from colloidal Fe-(oxyhydr)oxide particles has been called into question[64]. Fe measured by the pyrophosphate technique thus represents a potential maximum amount of organically complexed Fe. Briefly, 0.1 M Na$_4$P$_2$O$_7$·10H$_2$O (Alfa Aesar ACS grade #33385-22) was brought to pH 10 with Aristar Plus trace metal grade HCl (VWR chemicals #87003-259). 30 mL of the pyrophosphate extractant was added to 300 mg of 105 °C oven dried powdered bulk soil material, and the mixture was shaken overnight for 16 h at 180 rpm (Eberbach Mod. 6010 shaker table). The samples were then centrifuged at 15,550×$g$ (10,980 rpm) for 10 min, and the supernatant was decanted from the overtop of the soil pellet and filtered through a 0.2 μm filter (VWR #28145-499). Samples were analyzed for Fe within 5 days of the extraction procedure by atomic absorption spectroscopy on a Thermo iCE 3000 Flame Atomic Absorption Spectrometer. The sodium pyrophosphate results are presented in the Supplementary Data 2.

The hydroxylamine hydrochloride extraction to analyze Fe and Si within amorphous material was applied following the methodology of Shang and Zelazny (2008)[74], a modification of the methods used by Ross et al. [84] and Chao and Zhou (1983)[83]. The hydroxylamine hydrochloride technique acts as a combined reduction and acid protonation technique and is selective for Fe and Si-containing amorphous material[74,81–84]. The hydroxylamine hydrochloride technique was chosen rather than the ammonium oxalate in darkness (AOD) method as it extracts Fe in comparable amounts from amorphous material[74,84], because the hydroxylamine technique does not attack Fe within magnetite or chlorites and exhibits minimal potential for dissolving crystalline Fe-(oxyhydr)oxides such as lepidocrocite[74,83,84], and because the hydroxylamine supernatant produces minimal interference during absorbance measurements compared to the oxalate supernatant[74]. 25 mL of 0.25 M ACS grade NH$_2$OH·HCl (Beantown Chemical Catalogue #144935) + 0.25 M Aristar Plus trace metals grade HCl (VWR chemicals #87003-253) solution was added to 100 mg of 105 °C oven dried powdered bulk soil material from each sampled depth interval (Supplementary Data 3) within acid washed 50 mL centrifuge tubes. Samples were shaken for 16 h at 180 rpm at room temperature (Eberbach Mod. 6010 shaker table), and the supernatant was then filtered through a sterile 0.2 μm filter (VWR #28145-499). The filtered supernatant samples were acidified to 2% v/v with trace metals grade HNO$_3$ (VWR chemicals Aristar Plus #87003-259). The samples were then diluted 10× using a 2% v/v trace metals grade HNO$_3$ in 18.2 MΩ DI-water solution and analyzed for Fe and Si content via atomic absorption spectroscopy on a Thermo iCE 3000 Flame Atomic Absorption Spectrometer.

The citrate dithionite extraction was applied to examine Fe within both amorphous and crystalline (oxyhydr)oxides[74,89] based on the procedure of the 2004 Soil Survey Laboratory Manual[151]. The dithionite acts as a reducing agent for Fe$^{3+}$ in Fe$^{3+}$-containing amorphous and crystalline materials at near-neutral pH conditions[74,86–89]. Some work has suggested that in addition to the reduction of Fe from within oxides and oxyhydroxides reduction with dithionite extracts, minor amounts of Fe$^{3+}$ from vermiculite interlayers[156–158] and limited but measurable quantities of structural Fe from within smectites[74,86–88]. Briefly, 25 mL of 0.57 M Na$_3$C$_6$H$_5$O$_7$·2H$_2$O, 400 mg of

$Na_2S_2O_4$, and 750 mg of 105 °C oven-dried powdered bulk soil material from each sampled soil depth interval (Supplementary Data 3) were combined within acid-washed 50 mL centrifuge tubes and shaken overnight for 12–16 h at 180 rpm (Eberbach Mod. 6010 shaker table), then agitated vigorously by hand for approximately 30 s to ensure suspension of soil particles in solution and allowed to incubate without shaking overnight at room temperature for 12–16 h. Samples were then centrifuged at 4000 rpm (2057×$g$) for 15 minutes and the supernatant was decanted from the soil pellet and filtered through a 0.2 μm filter (VWR #28145-499). Samples were acidified to 2% v/v with trace metal grade Aristar Plus $HNO_3$ (VWR chemicals #87003-259), diluted 100× using a solution of 2% v/v trace metal grade $HNO_3$ and 18.2 MΩ DI water, and then analyzed for Fe content via atomic absorption spectroscopy on a Thermo iCE 3000 Flame Atomic Absorption Spectrometer.

## Data availability

All data used for figure and chart creation can be found in Supplementary Data 1 and 2 as well as at the following online repository: https://data.mendeley.com/datasets/gr2skw8k4s/1.

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

## Acknowledgements

We acknowledge the use of resources from the Nevada Plasma Facility Lab (NPFL) at the University of Nevada, Las Vegas. We also acknowledge the use of facilities within the Eyring Materials Center at Arizona State University, supported in part by NNCI-ECCS-1542160. We acknowledge funding from NASA EPSCoR (Grant No. 80NSSC20M0043), the Nevada Space Grant (Grant No. NNX15AI02H), the Geological Society of America, and the Clay Minerals Society. We also acknowledge funding from the UNLV Jack and Fay Ross family fellowship, Graduate and Professional Student Association, and TTDGRA programs. We acknowledge the tireless work of Benjamin Azua and Daniel Panduro during field sampling. Sample collection from Gros Morne National Park in Newfoundland, Canada was conducted with the permission of the park.

## Author contributions

Anthony Feldman formulated basic project ideas and methodology, procured funding, collected samples, performed analyses, curated and evaluated data, produced figures and tables, wrote the original draft, and edited the manuscript. Elisabeth Hausrath formulated project ideas and methodology, supervised the project, procured funding, assisted with sample collection and analysis, and assisted with manuscript revisions. Elizabeth Rampe assisted with project conceptualization and methodology, funding acquisition, data analysis, and manuscript revisions. Valerie Tu assisted with methodology, analyses, data evaluation, and manuscript revisions. Tanya Peretyazhko assisted with project ideas and methodology and manuscript revisions. Christopher DeFelice assisted with methodology, sample analysis, data evaluation, and manuscript revisions. Thomas Sharp assisted with methodology, sample analysis, data evaluation, and manuscript revisions.

## Competing interests

The authors declare no competing interests.
