## [Peer Review File · Communications Earth & Environment]

17th Oct 23

Dear Mr Feldman,

Please accept my apologies for the unexpected delay in sending a decision on your manuscript. Your manuscript titled "Fe-rich X-Ray Amorphous Material Records Past Climate and Persistence of Water on Mars" has now been seen by three reviewers, whose comments are appended below. You will see that they find your work of some potential interest. However, they have raised quite substantial concerns that must be addressed. In light of these comments, we cannot accept the manuscript for publication, at least in its current form.

We hope you will find the reviewers' comments useful as you decide how to proceed. Should you decide to address these criticisms, please ensure to provide additional soil data and justification for the methodology used in the study as pointed out by the second and third reviewers.

If you choose to re-submit your manuscript, please either highlight all changes in the manuscript text file or provide a list of the changes made to the manuscript along with your point-by-point response to the reviewers' concerns.

If the revision process takes significantly longer than three months, we will be happy to reconsider your paper at a later date, as long as nothing similar has been accepted for publication at Communications Earth & Environment or published elsewhere in the meantime.

Please use the following link to submit your revised manuscript, point-by-point response to the reviewers' comments with a list of your changes to the manuscript text (which should be in a separate document to any cover letter), a tracked-changes version of the manuscript (as a PDF file) and any completed checklist:

[link redacted]

Please do not hesitate to contact us if you have any questions or would like to discuss the required revisions further. Thank you for the opportunity to review your work.

Best regards,

Mojtaba Fakhraee, PhD

Editorial Board Member
Communications Earth & Environment
orcid.org/0000-0002-2461-6374

Joe Aslin
Senior Editor
Communications Earth & Environment

EDITORIAL POLICIES AND FORMAT

If you decide to resubmit your paper, please ensure that your manuscript complies with our editorial policies and complete and upload the checklist below as a Related Manuscript file type with the revised article:

Editorial Policy Policy requirements (Download the link to your computer as a PDF.)

For your information, you can find some guidance regarding format requirements summarized on the following checklist: (<https://www.nature.com/documents/commsj-phys-style-formatting-checklist-article.pdf>) and formatting guide (<https://www.nature.com/documents/commsj-phys-style-formatting-guide-accept.pdf>).

REVIEWER COMMENTS:

Reviewer #1 (Remarks to the Author):

Techniques commonly used in soil science were employed to study the X-ray amorphous material formation and longevity within terrestrial Fe-rich soils of different ages in terrestrial mediterranean, subarctic, and desert climates and further to understand the implications of this material for past aqueous and climatic conditions in Gale crater and elsewhere on Mars. Among them, the selective dissolution methods and the Rietveld refinements of powder XRD patterns were massively used, and many interesting results have been obtained. The discussion is sound and well supported by these results. Moreover, this work shows great potential of using these techniques in characterizing the future mission-returned martian samples. I am glad to recommend this nice work to be published after the following minor issues are addressed.

My minor comments:

Line 26-27. It is recommended to show what minerals might be present in the “amorphous” materials. You can refer to e.g., J. Geophys. Res.: Planet 2021, 126 (5), e2020JE006769.

Line 28. I think “hydrothermal alterations” are more proper than “serpentinization”. Some recent work can be cited, for example, J. Geophys. Res.: Planet 2020, 125 (12), e2020JE006590.

Line 222-223. As Co radiation is more suitable for studying Fe-rich samples, why was a Cu K α instrument utilized to study the bulk soil and oriented mount clay-size sample?

In figure 5, the legend can be improved. At present, A denote amorphous and B denote Fe-rich amorphous. It will be more logic that A+B denote amorphous and B denote Fe-rich amorphous. I think I make myself clear.

In the reference list, why do some refs (e.g., ref 9, 28, and 31) include preprint information? Additionally, the information of some refs (e.g., ref 32) is incomplete.

Reviewer #2 (Remarks to the Author):

- 1) The mayor clime of this work is to compare soils properties on Earth with sample data on Mars, and possibly draw general conclusion for Martian surface materials.
- 2) The specific questions are new to me and would be relevant if the comparison of data extracted from the materials on Earth and Mars are sufficiently well specified to be comparable.
- 3) The paper will be of relevance to scientists working in the Mars surface process area.
- 4) The research field of soil mineralogy will here look into relevant questions, which has been discussed for many decades in soil science. The relation to climate development I find irrelevant.
- 5) I number of detailed analysis methods have been used, but from a soil scientist point of view basic soils data has to be given much more detailed from the introduction to leave the reader with a soil science background a chance to find the paper relevant to continue reading. General levels of information is far too unspecific.
- 6) Soils on Earth all contain a number of horizons with different texture properties, and always some organic matter. There has been done extractions on the soil material, and it will be relevant from start to know what we have to do with. Natural grain sizes, or ground material as used on Mars. Extraction fractions, standard methods. All these data might not be available from Mars data, but we need them anyway. And though the amount of amorphous material is determined from the background calculated in the XRD refinement process this is not a precise method to use, though it was used on Mars. Chemical extractions and use of Mössbauer spectroscopy would give much more reliable results, which is possible for Earth Material.
- 7) I have no nearly complete overview of all the literature on data from Mars.
- 8) If the work should be relevant for publishing I think it should be considered taken up after the editors have considering my comments. There are many sophisticated methods used. However, basic soils data is still relevant. The development of the planets surface is a slow process, so basic soils data are still relevant in evaluating what we have to do with. I think that relevant and interesting questions are asked, but the treatment is so far too inaccessible.

Reviewer #3 (Remarks to the Author):

Revision of the paper "Fe-rich X-Ray Amorphous Material Records Past Climate and Persistence of Water on Mars" The authors investigated X-ray amorphous material formation and longevity within terrestrial Fe-rich soils of different ages in terrestrial mediterranean, subarctic, and desert climates using bulk and selective dissolution methods. The aim is to understand the implications of amorphous material for past aqueous and climatic conditions in Gale crater and elsewhere on Mars. To achieve the stated goals, the authors have conducted a large set of experiments using complementary analytical techniques for studies the Fe-rich X-Ray Amorphous Material. The author established that in situ aqueous alteration is required to concentrate Fe into the clay size material and to form abundant Fe-containing poorly ordered material. Therefore they predict that cooler climates promote the formation and persistence of Fe-rich X-ray amorphous material whereas warmer climates promote the formation of crystalline secondary phases. My overall feeling is that the paper has merit and should to be published with mayor revision, but the paper is overly technical and very difficult to follow. The technical nature of the subject is compounded by some

punishingly short paragraphs. For the "average" reader of Soil Mineralogist background the paper will be hard going. The strength of the paper is the thorough testing of various mineralogical methods on the crystallinity of X-ray amorphous material form, I think that the existence of all these poorly ordered materials proves to be confusing and have little or no foundation. While the analyses seem to have been carried out with care, the reader has a very hard time appreciating the importance of the findings and linking them to the climate persistence with past cool and relatively wet environments on Mars. There are several reasons for this, explained below.

Abstract is technical in nature. There is no clear evidence of the mineralogy Fe-rich X-ray amorphous material in Mars. I would also rewrite the sentences between lines 7 and 8. I find it too difficult to understand and to follow the explanation as "This X-ray amorphous material is variably siliceous, Fe-rich, and contains volatiles, and it therefore likely contains incipient weathering products". For soil mineralogist the Fe amorphous material are a function of crystallite disorder, smaller sizes and metals inclusions, that should be considered in assessing paleoenvironmental behaviors of ferrihydrite.

The Introduction focus on the Chemistry and Mineralogy (CheMin) instrument on the Curiosity rover at Gale crater, but no source has been found in the references for this quotation, some of them are random without any relations with the subject. The bibliographic citation not addressed at all the Fe amorphous material, ignoring all the important literatures of ferrihydrites, to connect the crystallite size with the physicochemical properties and paleoenvironmental behaviors, see DOI:

10.1021/acsearthspacechem.8b00179; DOI: 10.1006/jcis.1998.5899; DOI: 10.1016/0167-2738(88)90202-0; DOI: 10.1007/s003390101175; DOI: 10.1016/s0277-5387(00)80487-8; DOI: 10.1126/science.1142525 and in particular for the statements in L54-59, L121-162 please include this references DOI: 10.1029/2007jb005207; DOI: 10.1007/bf00356674; DOI: 10.1007/bf02538060; DOI: 10.1016/0016-7037(81)90250-7; DOI: 10.1007/978-94-009-4007-9

The discussion section in the lines 305-368 is based on hydroxylamine hydrochloride dissolution, which is not specific for iron poorly mineral but for Mn oxides minerals, see <https://doi.org/10.2136/sssaj1972.03615995003600050024x> [https://doi.org/10.1016/S1631-0713\(02\)01753-4](https://doi.org/10.1016/S1631-0713(02)01753-4). For example, the Figure 2 is confused, most statements could be reconsidered and connected with the Fe chemical-crystallographic and dissolution data. In general, the large experimental design is completely unsuitable to follow most explanations given in this section. I suggest the Authors to use the Fe dissolution treatments reported in Cornell, R.M., Schwertmann, U., 1996. The Iron Oxides: Structure, Properties, Reactions, Occurrences and Uses. VCH, Weinheim, Germany. X-ray data was not conclusive as to the identify the Fe amorphous minerals, for example Figure 2 – X-ray amorphous contributions to field soil XRD patterns. There are many research on differential XRD a particularly useful approach for identifying and characterizing poorly crystallized iron-oxide minerals in soil and concretions. The Differential X ray diffraction technique was useful to increase the detection limit of iron oxides in soils as well as a mean to obtain an XRD scans of the solid phase destroyed by a selective dissolution method. The XRD pattern resulting from the difference between the pattern with and the pattern without the solid phase (that is, the DXRD pattern) has to be subjected to adjustments to avoid negative peaks and other artifacts produced by the difference operation, see <https://doi.org/10.2136/sssaj1981.03615995004500020040x>. The second problem concern the "extent of crystallinity of Fe-containing phases under different climate conditions in soils of different ages." applied for the iron oxides, The authors also have to explain better the FeH/FeD index applied and related with Fe forms. I don't understand the statement at pag 11 (lines 7-13), but since part of the discussion was based on this index within studied soils I suggest the authors to apply the already widely accepted methods proposed in international literature (Schwertmann et al., 1982; Schwertmann and Cornell, 1991.) or to explain well why they use other methodology.

I am sorry for the Authors but not revision can improve this paper, I had a very difficult time understanding the ms. if the Authors try to reorganize this paper I can review it again.

2/13/2024

Responses to Reviewers

Reviewer: 1

Recommendation: Publish after minor revision.

Remarks to the Author:

Techniques commonly used in soil science were employed to study the X-ray amorphous material formation and longevity within terrestrial Fe-rich soils of different ages in terrestrial mediterranean, subarctic, and desert climates and further to understand the implications of this material for past aqueous and climatic conditions in Gale crater and elsewhere on Mars. Among them, the selective dissolution methods and the Rietveld refinements of powder XRD patterns were massively used, and many interesting results have been obtained. The discussion is sound and well supported by these results. Moreover, this work shows great potential of using these techniques in characterizing the future mission-returned martian samples. I am glad to recommend this nice work to be published after the following minor issues are addressed.

We appreciate the reviewer's interest and commentary and believe that the changes noted for the following points have improved the article.

Line 26-27. It is recommended to show what minerals might be present in the "amorphous" materials. You can refer to e.g., J. Geophys. Res.: Planet 2021, 126 (5), e2020JE006769.

We appreciate the reviewer's interest in clarifying the possible nanomineral content of the X-ray amorphous material. We have a separate manuscript currently under review where we detail a microscale investigation using extensive additional TEM and synchrotron microprobe data to evaluate the structure and chemistry of the nanomaterials in the X-ray amorphous fraction.

However, we do present a subset of TEM images in this manuscript where we see evidence for the potential presence of ferrihydrite and goethite nanoparticles (Figure 1). However as in Smith and Horgan (2021) none of the d-spacings gleaned from TEM diffractograms fully match with the d-spacings for known nanominerals. We have added d-spacings to the inset in panel A of Figure showing the presence of nanocrystallites and the following text from lines 200 to 203—

“D-spacings of $\sim 2.45\text{\AA}$ and 2.10\AA from nanocrystallites do not match ideal structures for known nanominerals but are consistent with the presence of ferrihydrite or goethite nanoparticles (Anthony et al., 2003).”

Line 28. I think "hydrothermal alterations" are more proper than "serpentinization". Some recent work can be cited, for example, J. Geophys. Res.: Planet 2020, 125 (12), e2020JE006590.

We appreciate the reviewer pointing out a way to clarify this statement and have instituted this change in what is now line 32. We have also incorporated the suggested article describing allophane precipitation from gel.

Line 222-223. As Co radiation is more suitable for studying Fe-rich samples, why was a Cu K α instrument utilized to study the bulk soil and oriented mount clay-size sample?

We appreciate the reviewer's note regarding varying utility of Co and Cu anodes in XRD analysis of Fe-rich samples. We had a readily available Cu K α instrument in house which was utilized for XRD analyses to which Rietveld refinement was not going to be applied, i.e. bulk soil samples and the oriented clay mounts. As noted, Co radiation is more suitable for studying Fe-rich samples. We specifically shipped out samples for analysis on a Co K α source instrument at Johnson Space Center that were intended for use in Rietveld refinement. While use of a Co K α source instrument for all XRD analysis would have been beneficial, we chose to use the in-house Cu K α for samples that were used purely for phase identification without subsequent quantification procedures.

In figure 5, the legend can be improved. At present, A denote amorphous and B denote Fe-rich amorphous. It will be more logic that A+B denote amorphous and B denote Fe-rich amorphous. I think I make myself clear.

We appreciate the reviewer's comments on improving the clarity of Figure 5. To improve clarity, we have denoted the amorphous material as Fe-rich or Other.

In the reference list, why do some refs (e.g., ref 9, 28, and 31) include preprint information? Additionally, the information of some refs (e.g., ref 32) is incomplete.

We appreciate the reviewer's input on references and have corrected the references to show the appropriate non-preprint information and to contain complete information. We appear to be having issues stemming from an unknown error in the Mendeley citation engine utilized as a reference generator. The papers listed as preprints are not in fact preprints. However, when we make any changes involving a reference in the main text the references bibliography resets and puts back in the incorrect pre-print notation. We have reviewed the references list to manually remove preprint status and review formatting.

Reviewer: 2

Recommendation: Publish after major revision.

Remarks to the Author:

1) The mayor clime of this work is to compare soils properties on Earth with sample data on Mars, and possibly draw general conclusion for Martian surface materials.

We appreciate the reviewer elucidating the central focus of the paper. We are heartened that the central story of the manuscript is clear.

2) The specific questions are new to me and would be relevant if the comparison of data extracted from the materials on Earth and Mars are sufficiently well specified to be comparable.

We completely understand the reviewer's concern that the data from Earth and Mars must be comparable. We utilized the X-ray diffraction measurements as a tool for directly comparing the amorphous material in the terrestrial soils with the amorphous material observed in XRD patterns from Gale crater samples (Figure 2). As the use of X-ray diffraction to quantify amorphous material presence is admittedly indirect, we directly confirmed the presence of Fe-rich X-ray amorphous material including both amorphous and nanocrystalline material using TEM analyses (Figure 1). The XRD data contained amorphous "humps" with similar location to those in Gale crater (Figure 2), while the TEM work confirmed that the amorphous material present was Fe/Si-rich and Al-poor, chemically like the material observed in Gale crater (Figure 1). The manuscript subsection *Relevance of the Studied Field Amorphous Material to Amorphous Material at Gale, Crater Mars* was included with the reviewer's noted interest in comparability between terrestrial and Martian materials in mind. In addition, as described below (comment 5), we have added new text to the introduction of the main manuscript to describe basic field site selection criteria and soils data supplementing the material describing the choice of field sites and basic soils data given in the methods, supplementary material, and appendices.

3) The paper will be of relevance to scientists working in the Mars surface process area.

We appreciate the reviewer's indication of the relevance of this work.

4) The research field of soil mineralogy will here look into relevant questions, which has been discussed for many decades in soil science. The relation to climate development I find irrelevant.

The role of climate as a soil forming factor has been well established going back to the genesis of the state factor model (Jenny, 1941) and proposed as a factor in soil organic variation as far back as the 18th century (De Saussure, 1796). The impact of environmental conditions on the formation of secondary minerals such as Fe-oxides (e.g., Cudennec & Lecerf, 2006; Schwertmann et al., 1999; Schwertmann & Murad, 1983) and clay minerals (e.g., Bishop et al., 2018; Istok & Harward, 1982; Klopogge et al., 1999; Mosser-Ruck et al., 2010; Petit et al., 2017) is well established.

Formation of amorphous material has similarly been attributed to environmental conditions providing a kinetic limiting factor preventing development of more thermodynamically stable crystalline phases (Ziegler et al., 2003).

This project was developed to investigate what the presence of Al-poor and Fe/Si-rich X-ray amorphous material within martian sediments at Gale crater might imply about past environmental conditions during and post-formation. I.e., does the presence of Al-poor and Fe/Si-rich amorphous material help identify temperature and precipitation conditions (climate) during water-rock interaction in Gale crater sediments and rocks. Climatic conditions are

inherently relevant to the central question of the article and to the formation of secondary products in soil systems. As such, the climatic question is of necessity the central focus of the article because we are interested in the impact of different temperature and precipitation conditions on the formation and longevity of Fe-containing amorphous material found in both terrestrial and martian settings.

5) I number of detailed analysis methods have been used, but from a soil scientist point of view basic soils data has to be given much more detailed from the introduction to leave the reader with a soil science background a chance to find the paper relevant to continue reading. General levels of information is far too unspecific.

We appreciate the reviewers' desire for an expanded description of the soils investigated in this study. While soil mineralogical and chemical data have been presented in the manuscript under the subsections "Soil Mineral Inputs to the Amorphous Material-Forming Weathering System" and "Relevance of the Studied Field Amorphous Material to Amorphous Material at Gale, Crater Mars", we have incorporated 2 new sections of the article titled "Field Site Selection" and "Characteristics of the Soil System: Soil Morphology, pH, and Organic Content" to provide detail in the main text on the field sites and basic soils data. The new text for these sections is found on lines 62 to 154—

"Field Site Selection

Serpentine soils provide a useful analog setting to examine processes in Fe-rich and Al-poor settings that are chemically relevant to the amorphous material and secondary minerals present in Gale crater. Soils derived from such hydrothermally altered ultramafic rock are typically Mg/Fe/Si-rich and Al-poor (Alexander et al., 2006, 2007a; Alexander, 2009; D'Amico et al., 2015; Mayhew and Ellison, 2020), chemically similar to the Al-poor and variably Fe/Si-rich chemical composition of the Gale crater X-ray amorphous component (Rampe et al., 2020a; Thorpe et al., 2022). Fe-oxides and smectite clay minerals comprise the majority of secondary products in serpentine soils (Wildman et al., 1971; Istok and Harward, 1982; Alexander et al., 1990, 2007b; Bonifacio et al., 1997; Lee et al., 2003; Caillaud et al., 2004; Gleeson et al., 2004; Gaudin et al., 2011; Butt and Cluzel, 2013). Likely secondary amorphous material has also previously been postulated in such settings (Istok and Harward, 1982). As a result of these chemical and mineralogical similarities to many martian materials, serpentine laterites (Gaudin et al., 2011) and serpentinite rock (Tutolo et al., 2019; Grosch et al., 2021; Tutolo and Tosca, 2023) have previously been investigated as analog environments for aqueous alteration processes in Fe-rich and Al-poor martian systems. Partially serpentinitized lake sediments have similarly been utilized as analog environments for analyzing alteration patterns in martian paleolake settings with primarily mafic mineral sources (Goudge et al., 2017; Sheppard et al., 2019).

Four serpentinite bodies in three different climate zones were chosen for investigation to examine climatic influences on the production and persistence of Fe-containing amorphous material. These sites include the Trinity Ultramafic Body and Rattlesnake Creek Terrane within the Klamath Mountains of northern California, the Tablelands of Gros Morne National Park in

Newfoundland, Canada, and a serpentinite body near the ghost town of Pickhandle Gulch in western Nevada (Table S1, Figure S1). The modern climate in the Klamath Mountains is mediterranean, featuring hot and dry summers and cold and wet winters. NOAA 1991-2020 climate normals for the Weaverville, CA weather station (Station ID: USC00049490), near the String Bean Creek sampling site in the Rattlesnake Creek Terrane, record mean annual temperatures of 12.8°C, with a mean of 23.2°C in July and 3.7°C in December, and an annual average precipitation of ~100 cm (Skinner et al., 2006; Arguez et al., 2012), predominantly snowfall. NOAA 1981-2010 climate normals for the Sawyers Bar Ranger Station (Station ID: USC00048025), north of Trinity Lake and within the Trinity Alps, show a mean annual temperature of 12.8°C, with a mean of 22.9°C in July and 4.4°C in December, and an annual average precipitation of ~118 cm falling mostly as snow (Skinner et al., 2006; Arguez et al., 2012); 1991-2020 climate normals were not available for that site. The exact climate at the Eunice Bluff and Deadfall lake sites in the Trinity region is likely slightly colder due to a higher elevation (~2100 m) compared to ~600-650 m elevation for the monitoring stations. Similarly high-elevation regions of the Klamath Mountains have mean summertime temperatures (July-October) of ~12.7°C and mean wintertime (November-May) temperatures hovering around freezing (~0.9°C) though with substantial periods of below-freezing conditions and abundant snowfall (Garwood and Welsh, 2007). A dispersed coniferous forest covers much of the Klamath Mountains (Mohr et al., 2000; Skinner et al., 2006) and is present at all sampling locations examined within this study. The modern climate of the Tablelands is subarctic; daily mean temperatures at the nearby Cow Head weather station (Station ID: 8401335) maintained by Environment Canada range from -7.0 °C in January to 15.7 °C in August with a mean yearly temperature of 3.9 °C, with a yearly average precipitation of 120.4 cm distributed relatively evenly throughout the year as both snowfall and rain (Environment Canada, 2010). The Tablelands is relatively barren with occasional ground hugging shrubs and stunted pine trees (Bouchard et al., 1986; Brouillet et al., 1998). Pickhandle Gulch possesses a desert climate, with the nearest NOAA climate station to Pickhandle Gulch, in Mina, Nevada (Station ID: USC00265168), recording 1991-2020 climate normals of a 14.4 °C mean annual temperature, ranging from 2.7 °C in December to 27.9 °C in July, with an average precipitation of 14.2 cm annually (Arguez et al., 2012). Vegetation at Pickhandle Gulch is a sparse sagebrush cover typical of the Great Basin region (West, 1983).

Characteristics of the Soil System: Soil Morphology, pH, and Organic Content

Although the Klamath Mountains and Tablelands soils possess similar age ranges, soils exhibit greater visual indications of in-situ pedogenic alteration within the Klamath Mountains than at the Tablelands or at Pickhandle Gulch - see soil descriptions and associated pH and loss on ignition (LOI) values in Appendix C in the supplementary online material. As soil parent material and soil ages are similar in the Klamath Mountains and Tablelands, we attribute variations in soil development to climatically induced differences in in-situ aqueous alteration. Here we provide background on the soil environment the amorphous material is forming in.

Klamath Mountain soils exhibit a clear age progression. Munsell color hue varies from 10YR to 7.5YR in the youngest Klamath Mountain soils (Eunice Bluff and Deadfall Lake, ~12.1 ka) with 5YR hues observed in the oldest Klamath Mountain soil (String Bean Creek, undated). These increases in soil reddening are consistent with the formation of ferrihydrite and goethite, and potentially hematite as soil age increases (Torrent et al., 1983; Schwertmann, 1993). Similarly, clay-films are absent from the youngest Klamath Mountain soils but progressively become more visible as soil age increases, with clay films observed throughout the entire soil profile in the oldest Klamath Mountain soil except for in the surface A horizon. The increase in clay-film visibility with soil age is consistent with formation of secondary clay minerals. Soil peds are likewise absent or present as < 5mm diameter sub-angular blocky shapes in the youngest Klamath Mountain soils with sub-angular blocky peds up to 5cm in diameter present in the oldest Klamath Mountain soil. Klamath Mountain soil textures include loamy sands, sandy loams, and sandy clay loams.

Tablelands soils are relatively uniform in appearance regardless of soil age. Munsell hue varies from 2.5Y in the younger soils (~13 to 17.6 ka) to exclusively 10YR in the oldest (Trout River Gulch, >20 ka), consistent with formation of Fe-oxides such as ferrihydrite and goethite (Schwertmann, 1993). Indications for ped formation or clay-film development were not observed in any Tablelands soil. Tablelands soil textures include loamy sands, sandy loams, and sandy clay loams. Coincident with the relative lack of variation with age, Tablelands soils do not possess visual indications for soil horizon differentiation. Such lack of soil horizon differentiation is potentially attributable to freeze-thaw cycling and frost-heaving mixing soil material and preventing horizon formation, as has been previously observed in soils on the Tablelands (Roberts, 1980).

Pickhandle Gulch soils do not exhibit significant variation. Soils are thin (<10 cm), likely reflecting a predominantly detrital buildup with limited contributions from underlying bedrock. All soils exhibit Munsell colors with 10YR hues, a chroma of 4, and values of 2 or 3. Soils either lack ped development or possesses weakly developed sub-angular blocky peds < 5mm in diameter. Clay-films are absent from peds when present. Soil texture encompasses loamy sands, sandy loams, sandy clay loams, and sandy clays. Formation of soil pores at the surface of each soil profile is consistent with development of vesicular horizons through dust influx and sporadic precipitation (Turk and Graham, 2011; Dietze et al., 2012).

Soil pH and organic content determined by LOI likewise vary between field sites. Soil pH is slightly acidic to slightly alkaline (5.88 – 7.56) in the Klamath Mountains, slightly alkaline (7.30 – 8.30) in the Tablelands, and alkaline at Pickhandle Gulch (8.17 – 8.72). LOI values are greatest near the surface of Klamath Mountain soils, varying from as much as 21.97 wt.% at the surface to between 1.67 – 4.69 wt.% at depth. The vertical variation in LOI with a relatively monotonic decrease with depth likely reflects heterogenous distribution of organics within the soil profile resulting from contributions of litter from the overlying dispersed coniferous forest vegetation. LOI values in the Tablelands (6.24 – 11.06 wt.%) and at Pickhandle Gulch (1.03 – 2.32 wt.%) are both substantially lower than the maximum's observed in the Klamath Mountains and exhibit limited vertical variability, reflecting the lack of vegetation cover at both field sites and a

corresponding lack of direct leaf/plant litter contributing to organic enrichment at the soil surface. While LOI error is not exact, it can be estimated at roughly $\pm 2\%$ (Heiri et al., 2001). Variations in organic content are reflected in varying organically bound Fe-content as measured by a Na-pyrophosphate extraction (McKeague, 1967; Shang and Zelazny, 2008) (Figure S53; Appendix B). While Fe_P is minimal in all examined soils (≤ 1.16 mg Fe / g soil), relatively higher Fe_P in the Klamath Mountains (0.11 – 1.16 mg Fe / g soil) than Tablelands (0.02 – 0.30 mg Fe / g soil) or Pickhandle Gulch (0.00 – 0.03 mg Fe / g soil) soils is consistent with the relative contributions of organics as measured by LOI.”

6) Soils on Earth all contain a number of horizons with different texture properties, and always some organic matter. There has been done extractions on the soil material, and it will be relevant from start to know what we have to do with. Natural grain sizes, or ground material as used on Mars. Extraction fractions, standard methods. All these data might not be available from Mars data, but we need them anyway. And though the amount of amorphous material is determined from the background calculated in the XRD refinement process this is not a precise method to use, though it was used on Mars. Chemical extractions and use of Mössbauer spectroscopy would give much more reliable results, which is possible for Earth Material.

We appreciate the reviewer’s interest in clarity for the methodology used for our analyses. The descriptions of the methods, grain sizes used, and extraction fractions can all be found in the online methods section given after the main text of the manuscript. We describe the use of the fine earth (<2 mm diameter) size fraction for the selective chemical analyses, and the use of the clay-size fraction (<2 μ m diameter) and powdered parent material for the bulk chemical analyses within the included methods section. Throughout the paper when referring to specific chemical, XRD, or TEM analyses we state the use of the bulk soil or clay-size fraction. We have made sure to specify grain diameter at the first mention of each size fraction (line 164 for bulk soil and line 165 for clay-size fraction). The CheMin instrument uses a sieve to acquire <150 μ m diameter material for analysis (Blake et al., 2012; Rampe et al., 2020a).

We agree that Mossbauer spectroscopy would be useful as an additional method for quantifying the presence of amorphous material. We chose to use XRD analytical methods to allow for a direct comparison with similar XRD data from Gale crater samples. These XRD results semi-quantitatively demonstrated variation in amorphous abundance between different field environments (Figure 2). We also directly confirm the presence of amorphous and nanocrystalline material using transmission electron microscopy (Figure 1). In alignment with the reviewer’s statement that chemical extractions will produce more reliable results, selective chemical dissolution techniques (hydroxylamine hydrochloride and citrate dithionite) were employed as the primary method used in this paper to examine the abundances of Fe-containing amorphous material in each terrestrial soil (Figure 4). These selective chemical extractions (Figure 4) definitively demonstrated variations in amorphous abundance between field sites. As such, while Mossbauer analyses would provide further confirmation of the presence of Fe-containing amorphous material in our samples, we have already confirmed and analyzed the

variable abundance and crystallinity of such material using XRD, TEM, and selective chemical dissolutions.

7) *I have no nearly complete overview of all the literature on data from Mars.*

We appreciate the reviewer's interest in an overview of literature pertaining to amorphous material on Mars. The paper originally included references to a complete overview of the mineralogical results from the CheMin XRD instrument in Gale crater through the Glenn Torridon campaign (Rampe et al., 2020a; Thorpe et al., 2022). We have updated the references on lines 23 and 25 to include the recently published results for the Canaima drill hole (Chipera et al., 2023). We have also provided references for the APXS chemical data (Leshin et al., 2013; Sutter et al., 2017), though the references given for the mineralogical reviews include such chemical data already (Rampe et al., 2020a; Thorpe et al., 2022). We have also updated the list of references originally given for other instances of X-ray amorphous material exposures on Mars to give a more comprehensive listing of the presence of allophane, ferrihydrite and other Fe-oxides, opal and amorphous silica, and Fe-containing amorphous silicates on line 28 (Milliken et al., 2008; Poulet et al., 2008; Horgan and Bell, 2012; Rampe et al., 2012a; Morris et al., 2013; Weitz et al., 2013, 2014; Pan and Ehlmann, 2014; Bishop and Rampe, 2016; Ruff and Hamilton, 2017; Weitz and Bishop, 2019). The complete text for the introductory paragraph describing amorphous material presence in Gale crater and elsewhere on Mars has been expanded and now reads from lines 21 to 37 as—

“X-ray diffraction (XRD) measurements by the Chemistry and Mineralogy (CheMin) instrument on the *Curiosity* rover at Gale crater, Mars have demonstrated that X-ray amorphous material makes up 15% to 73 wt.% of all rock and eolian sediment samples analyzed to date (Rampe et al., 2020a; Thorpe et al., 2022; Chipera et al., 2023). This material has been shown to be variably Fe-rich (3.0 – 43.1 wt.% FeO_T) and Si-rich (3.6 – 75.9 wt.% SiO₂) and typically Al-poor (≤ 10.46 wt.% Al₂O₃) (Rampe et al., 2020a; Smith et al., 2021; Thorpe et al., 2022; Chipera et al., 2023), consistent with principally mafic sources (Siebach et al., 2017; Bedford et al., 2019). This material is hereafter referred to as “amorphous” although it likely encompasses amorphous and short-range order materials and nanocrystallites (Klug and Alexander, 1974; Smith et al., 2018; Holder and Schaak, 2019; Smith and Horgan, 2021). Despite the prevalence of amorphous material in Gale crater (Rampe et al., 2020a; Smith et al., 2021; Thorpe et al., 2022; Chipera et al., 2023) and elsewhere on Mars (Milliken et al., 2008; Poulet et al., 2008; Horgan and Bell, 2012; Rampe et al., 2012b; Morris et al., 2013; Weitz et al., 2013, 2014; Bishop and Rampe, 2016; Ruff and Hamilton, 2017; Weitz and Bishop, 2019), the implications of its presence for past environmental conditions on Mars remain poorly understood (Smith et al., 2021). The Gale crater amorphous component could represent in-situ pedogenic or diagenetic alteration products including alteration products or chemical precipitates or detrital influx of glasses from dust or fluvial inputs⁴. While amorphous material can form from a variety of processes including as glass from melt and through hydrothermal alteration processes (Tutolo et al., 2019; Du et al., 2020), the presence of volatiles (e.g., H₂O, CO₂, SO₂) measured by the Sample Analysis at Mars (SAM) instrument suite suggests at least some of this material likely consists of secondary

weathering products (Leshin et al., 2013; Sutter et al., 2017; Rampe et al., 2020a; Smith et al., 2021). As amorphous material represents a substantial component of the Gale crater sedimentary rocks and aeolian sediments and is present across the Martian surface, elucidating the processes controlling formation and preservation of this material is critical for understanding sedimentation and diagenetic processes on Mars (McLennan et al., 2019; Smith et al., 2021).”

8) If the work should be relevant for publishing I think it should be considered taken up after the editors have considering my comments. There are many sophisticated methods used. However, basic soils data is still relevant. The development of the planets surface is a slow process, so basic soils data are still relevant in evaluating what we have to do with. I think that relevant and interesting questions are asked, but the treatment is so far too inaccessible.

We appreciate the reviewer’s interest in the methods and conclusions of the article. We also appreciate the reviewer’s interest in giving a broader overview of basic soils data from the examined field sites within the main text. As described above, we have included new sections on field site selection and soil morphology within the main manuscript text and expanded the discussion of the presence of amorphous material at Gale crater in the introduction.

Reviewer: 3

Recommendation: Publish after major revisions

Remarks to the Author:

The authors investigated X-ray amorphous material formation and longevity within terrestrial Fe-rich soils of different ages in terrestrial mediterranean, subarctic, and desert climates using bulk and selective dissolution methods. The aim is to understand the implications of amorphous material for past aqueous and climatic conditions in Gale crater and elsewhere on Mars. To achieve the stated goals, the authors have conducted a large set of experiments using complementary analytical techniques for studies the Fe-rich X-Ray Amorphous Material. The author established that in situ aqueous alteration is required to concentrate Fe into the clay size material and to form abundant Fe-containing poorly ordered material. Therefore they predict that cooler climates promote the formation and persistence of Fe-rich X-ray amorphous material whereas warmer climates promote the formation of crystalline secondary phases. My overall feeling is that the paper has merit and should to be published with mayor revision, but the paper is overly technical and very difficult to follow. The technical nature of the subject is compounded by some punishingly short paragraphs. For the "average" reader of Soil Mineralogist background the paper will be hard going. The strength of the paper is the thorough testing of various mineralogical methods on the crystallinity of X-ray amorphous material form, I think that the existence of all these poorly ordered materials proves to be confusing and have little or no foundation. While the analyses seem to have been carried out with care, the reader has a very

hard time appreciating the importance of the findings and linking them to the climate persistence with past cool and relatively wet environments on Mars. There are several reasons for this, explained below.

We appreciate the reviewer's commentary regarding the merit of the paper as well as in clarifying the presence of X-ray amorphous material and the writing overall. In response, we have rewritten it substantially to make the technical material more accessible and easier to follow, removing specifically the short paragraphs, which can be seen in the tracked changes manuscript. We have expanded the paper to now incorporate new sections regarding field site selection and basic soil morphology data. As regards the existence of the X-ray amorphous material, TEM observations (Figure 1) directly confirm the presence of both truly amorphous material completely lacking crystalline structure, as well as the presence of nanocrystallites with d-spacings suggestive of the presence of goethite. XRD is utilized in a manner like that employed by the Mars Science Laboratory CheMin team for analyzing the contribution of amorphous phases to the Gale crater XRD samples in order to present relative abundances. As well, we utilize standard soil selective chemical dissolution techniques designed to attack amorphous to poorly ordered phases (hydroxylamine hydrochloride) and Fe-oxides (citrate dithionite) which similarly suggest variations in the presence of amorphous Fe-phases between soils developing under different climate conditions. We answer in more detail the questions and concerns in the following responses.

Abstract is technical in nature. There is no clear evidence of the mineralogy Fe-rich X-ray amorphous material in Mars. I would also rewrite the sentences between lines 7 and 8. I find it too difficult to understand and to follow the explanation as "This X-ray amorphous material is variably siliceous, Fe-rich, and contains volatiles, and it therefore likely contains incipient weathering products".

We appreciate the reviewer's interest in whether Fe-rich X-ray amorphous material exists on Mars. There is a plethora of evidence available for the existence of X-ray amorphous material in Martian mineral assemblages, and Fe-rich X-ray amorphous material in particular. As noted elsewhere in this review, ferrihydrite represents a common nanocrystalline (Eggleton and Fitzpatrick, 1988; Drits et al., 1993; Jansen et al., 2002; Michel et al., 2007; Rancourt and Meunier, 2008; Manceau, 2009; Cismasu et al., 2011) and thus X-ray amorphous (Holder and Schaak, 2019; Smith and Horgan, 2021) constituent of terrestrial soils (e.g., Parfitt et al., 1988; Stucki et al., 1988; Childs, 1992; Schwertmann, 1993; Cismasu et al., 2011; Jiang et al., 2018). Nanocrystalline Fe-oxides including ferrihydrite and hematite have been directly documented in Martian sediments and rocks (Chevrier and Mathé, 2007; Ruff and Hamilton, 2017; Rampe et al., 2020b). Similarly, amorphous and opaline silica has been documented within Martian sediments and rocks (Milliken et al., 2008; Ruff et al., 2011; Rapin et al., 2018; Sun and Milliken, 2018). Allophane (Rampe et al., 2012a) and amorphous Fe-containing silicates (Weitz et al., 2014) have also been directly documented in Martian materials.

Within Gale crater the contribution of the X-ray amorphous component to each examined XRD samples has been modelled using the FULLPAT program (Chipera and Bish, 2002). These fits show that X-ray amorphous material, as noted in the manuscript in lines 24 and 25, make up between 15 to 73 wt.% of each sample while mass balance calculations using the FULLPAT fits combined with bulk chemical data from the Alpha Particle X-Ray Spectrometer (APXS) instrument suggest the chemical content of the X-ray amorphous fraction is 3.0 – 43.1 wt.% FeO and 3.6 – 75.9 wt.% SiO₂ (Rampe et al., 2020a, 2020b; Thorpe et al., 2022; Chipera et al., 2023). The volatile content described above comes from evolved gas analysis using the Sample Analysis at Mars (SAM) instrument which detects and identifies organic molecules. The presence of such volatiles is indicative of secondary development of material rather than pure inheritance of primary glassy material. The presence of secondary amorphous material is similarly indicated by mass balance and mixing line calculations (Smith et al., 2021).

We have changed lines 7 and 8 in the abstract, now from lines 6 to 8, to clarify the meaning as follows—

“This X-ray amorphous material is variably siliceous and Fe-rich but Al-poor (Rampe et al., 2020a; Thorpe et al., 2022; Chipera et al., 2023). The presence of volatiles is consistent with the presence of incipient weathering products (Leshin et al., 2013; Sutter et al., 2017; Rampe et al., 2020a; Smith et al., 2021; Thorpe et al., 2022).”

The references provided above are not given in the abstract itself but are cited with the following similar statements in the first paragraph of the introduction.

Lines 23 to 25: “This material has been shown to be variably Fe-rich (3.0 – 43.1 wt.% FeO_T) and Si-rich (3.6 – 75.9 wt.% SiO₂) and typically Al-poor (≤ 10.46 wt.% Al₂O₃) (Rampe et al., 2020a; Smith et al., 2021; Thorpe et al., 2022; Chipera et al., 2023), ...”

Lines 32 to 34: “... the presence of volatiles (e.g., H₂O, CO₂, SO₂) measured by the Sample Analysis at Mars (SAM) instrument suite suggests at least some of this material likely consists of secondary weathering products (Leshin et al., 2013; Sutter et al., 2017; Rampe et al., 2020a; Smith et al., 2021).”

For soil mineralogist the Fe amorphous material are a function of crystallite disorder, smaller sizes and metals inclusions, that should be considered in assessing paleoenvironmental behaviors of ferrihydrite.

The reviewer is correct that the X-ray amorphous component includes both truly amorphous gels, nanocrystallites, short-range order material, and poorly-crystalline phases (Klug and Alexander, 1974; Holder and Schaak, 2019; Smith and Horgan, 2021; Rampe et al., 2022) and that such trends are affected by metal inclusions. We are interested, within this study, in studying the bulk trends in amorphous abundance and crystallinity. As well, and as noted in a response below, ferrihydrite is far from the only potential Fe-containing X-ray amorphous material present in the examined soils. The X-ray amorphous component of a complex soil system is likely to

incorporate a plethora of different constituents, many of which do not match the chemistry or structure of known nanominerals such as ferrihydrite (e.g., Istok and Harward, 1982; Smith and Horgan, 2021; Rampe et al., 2022). In that vein, we confirm the presence of such amorphous and nanocrystalline material and the potential presence of ferrihydrite and goethite nanocrystallites using transmission electron microscopy (Figure 1) and demonstrate that the amorphous contribution is consistent with the contribution to Gale crater XRD patterns (Figure 2). However, the goal of the XRD analyses is to confirm the relevance of our terrestrial samples to the Gale crater X-ray amorphous material and to note the bulk mineralogy of the fine earth and clay-size fractions. The central story of the paper then focuses on the selective dissolution methods to quantify changes in abundance of Fe-containing amorphous material over time between the different climatic conditions. Thus, a microscale investigation of nanominerals present within our samples is outside the scope of this paper and is detailed in a separate manuscript.

The Introduction focus on the Chemistry and Mineralogy (CheMin) instrument on the Curiosity rover at Gale crater, but no source has been found in the references for this quotation, some of them are random without any relations with the subject.

We appreciate the reviewer's request for clarity in regard to the CheMin instrument but respectfully are confused by and disagree with their assertion that no references, and indeed random references, are given for the CheMin instrument results. Is the reviewer requesting the specific references that indicates that amorphous material ranges in percentage from 15-73%? That number results from all of the measurements performed to date, which are documented in many papers. For a single source to compare all the samples, we have also added the online CheMin data base here: <https://odr.io/chemin#/search/display/84/eyJkdF9pZCI6IjQzIn0/1>

The references given in line 23 include a review of the complete CheMin XRD data and APXS chemical data prior to the Glen Torridon campaign (Rampe et al., 2020a), a complete overview of such results from the Glen Torridon campaign (Thorpe et al., 2022), and we have now added the most recently published CheMin analysis of the Canaima drill hole (Chipera et al., 2023). We also provide a reference discussing the chemical variability of the Gale crater amorphous material and resulting implications for formation processes (Smith et al., 2021). While there are a plethora of references that directly describe each individual CheMin borehole XRD sample, the provided references cover all CheMin XRD and associated APXS data published to date, with all of the samples collated in the online database. Respectfully, we do not understand the origin of this critique and find it to not accurately describe the content of the manuscript.

The bibliographic citation not addressed at all the Fe amorphous material, ignoring all the important literatures of ferrihydrites, to connect the crystallite size with the physicochemical properties and paleoenvironmental behaviors, see DOI: 10.1021/acsearthspacechem.8b00179; DOI: 10.1006/jcis.1998.5899; DOI: 10.1016/0167-2738(88)90202-0; DOI: 10.1007/s003390101175; DOI: 10.1016/s0277-5387(00)80487-8; DOI: 10.1126/science.1142525 and in particular for the statements in L54-59, L121-162 please

include this references DOI: 10.1029/2007jb005207; DOI: 10.1007/bf00356674; DOI: 10.1007/bf02538060; DOI: 10.1016/0016-7037(81)90250-7; DOI: 10.1007/978-94-009-4007-9.

We appreciate the reviewer's desire to incorporate a larger focus on ferrihydrite into the article. While ferrihydrite is an important nanocrystallite (Michel et al., 2007) that would typically fall into the X-ray amorphous category based on crystallite size (e.g., Klug and Alexander, 1974; Holder and Schaak, 2019), it does not represent the only potential nanocrystalline, poorly-crystalline, or amorphous material present in the examined terrestrial soils. A plethora of other amorphous gels (Istok and Harward, 1982), short-range order phases such as hisingerite (Kohyama and Sudo, 1975; Farmer, 1992; Eggleton and Tilley, 1998) or allophane (Wada et al., 1972; Parfitt et al., 1988; Parfitt and Kimble, 1989; Du et al., 2020), as well as amorphous silica or opaline material (Gutiérrez-Castorena et al., 2005; Alfredsson et al., 2015; Drees et al., 2018) might be present. We see indications for the presence of goethite or potentially ferrihydrite in TEM images of X-ray amorphous material (Figure 1). We also see indications for the presence of nano to poorly crystalline goethite in the clay-size fraction of the Klamath Mountain and Tablelands soils (Figures S6, S9, S12, S15, S18, S21). For a detailed discussion of the nanominerals and amorphous gels present in these soils we would direct the reviewer to our separate preprint (Feldman et al., 2023). In the current manuscript, we first used XRD to establish the examined terrestrial amorphous material as relevant to Gale crater material (Figures 1 and 2). Subsequently, we used selective dissolution techniques to examine trends in secondary Fe crystallinity between field sites and over time within each field site (Figure 4). As such, a focus on the specific nanocrystallites present was touched on briefly (Figure 1) but is overall beyond the scope of this manuscript.

The discussion section in the lines 305-368 is based on hydroxylamine hydrochloride dissolution, which is not specific for iron poorly mineral but for Mn oxides minerals, see <https://doi.org/10.2136/sssaj1972.03615995003600050024x> [https://doi.org/10.1016/S1631-0713\(02\)01753-4](https://doi.org/10.1016/S1631-0713(02)01753-4).

We appreciate the reviewer's interest in the utility of the hydroxylamine hydrochloride extraction for determining amorphous Fe content but respectfully disagree with their statement that the dissolution method is not specific for amorphous, short-range order, and poorly crystalline Fe-oxides. The reviewer is correct that hydroxylamine hydrochloride has been utilized as a selective dissolution technique for Mn-oxides (e.g., Chao, 1972). However, a modified form of the hydroxylamine dissolution technique selective for dissolution of amorphous and poorly crystalline Fe-oxides in a manner that gives similar results to the commonly utilized ammonium oxalate in darkness (AoD) technique has been in use for decades (Chester and Hughes, 1967; Chao and Zhou, 1983; Ross et al., 1985; McAlister and Smith, 1999; Shang and Zelazny, 2008; Kiczka et al., 2011). The USDA NRCS methodology for amorphous to poorly crystalline Fe-oxide dissolution also utilizes a hydroxylamine hydrochloride technique (Soil Survey Staff, 2004). We chose to use the Hydroxylamine dissolution technique due to the aforementioned similar results to AoD and the significantly easier and less laboratory intensive procedure.

Hydroxylamine has similarly been used previously to extract amorphous Fe-oxides within a sequential extraction (Kiczka et al., 2011). We detail the hydroxylamine procedure in the methods section but also refer the reader to the section on selective dissolution techniques contained in the most recent Soil Science Society of America Soil Mineralogy Textbook (Shang and Zelazny, 2008).

For example, the Figure 2 is confused, most statements could be reconsidered and connected with the Fe chemical-crystallographic and dissolution data. In general, the large experimental design is completely unsuitable to follow most explanations given in this section. I suggest the Authors to use the Fe dissolution treatments reported in Cornell, R.M., Schwertmann, U., 1996. The Iron Oxides: Structure, Properties, Reactions, Occurrences and Uses. VCH, Weinheim, Germany. X-ray data was not conclusive as to the identify the Fe amorphous minerals, for example Figure 2 – X-ray amorphous contributions to field soil XRD patterns. There are many research on differential XRD a particularly useful approach for identifying and characterizing poorly crystallized iron-oxide minerals in soil and concretions. The Differential X ray diffraction technique was useful to increase the detection limit of iron oxides in soils as well as a mean to obtain an XRD scans of the solid phase destroyed by a selective dissolution method. The XRD pattern resulting from the difference between the pattern with and the pattern without the solid phase (that is, the DXRD pattern) has to be subjected to adjustments to avoid negative peaks and other artifacts produced by the difference operation, see <https://doi.org/10.2136/sssaj1981.03615995004500020040x>.

We appreciate the reviewer's interest in clarification of our choice of methodology for examining Fe-containing X-ray amorphous material. While we do have a second paper currently under review that examines the nanoscale mineralogy and chemistry of the amorphous material and nanocrystalline phases present in the X-ray amorphous component (Feldman et al., 2023), the focus of this paper is on bulk scale amorphous Fe content and the overall crystallinity of Fe-containing secondary phases, not the particular Fe-oxide containing nanominerals present within the soil system. However, we agree that characterizing such material is important. To that end we applied transmission electron microscopy (TEM) to the clay-size fraction. TEM provides a substantially more detailed analysis of the X-ray amorphous component than could reasonably be acquired using bulk XRD techniques (Smith and Horgan, 2021). TEM shows directly the presence of truly amorphous material as well as nanocrystallites with d-spacings consistent with the presence of the Fe-oxide goethite and/or ferrihydrite (Figure 1). Such findings are explored in more detail in the previously referenced pre-print (Feldman et al., 2023). Also, as noted in table 1 and extensively in the supporting online information, there are peaks in most clay-size fraction XRD patterns indicative of the presence of goethite (see supporting online information figures S6, S9, S12, S15, S18, S21). Thus, while application of XRD prior to and post application of the hydroxylamine and dithionite selective dissolutions could provide additional confirmation of the presence of iron oxides within the soil system, the presence of such phases has already been reasonably established in both crystalline (with XRD) and nanocrystalline (with TEM) forms.

Figure 2 was specifically designed to compare amorphous material within our terrestrial samples and amorphous material in the Gale crater samples using XRD techniques available on the Curiosity rover. While we directly confirmed the presence of amorphous material in our samples using TEM, it is important to confirm the presence of such material and ascertain its similarity to martian samples using techniques also available in-situ at Gale crater. As such, we designed the analytical techniques and layout for figure 2 to use the kinds of XRD analyses that are employed for the Curiosity rover CheMin data. We acknowledge that more intensive techniques are available for XRD analyses on terrestrial samples. However, we have already employed TEM analyses to confirm the presence and investigate the nature of amorphous and nanocrystalline materials in our samples. We have also confirmed the presence of XRD peaks attributed to goethite and ferrihydrite in our samples. As such, Figure 2 was explicitly designed to directly compare our material with martian materials using techniques similar to those used when analyzing gale crater CheMin samples.

The second problem concern the “extent of crystallinity of Fe-containing phases under different climate conditions in soils of different ages.” applied for the iron oxides, The authors also have to explain better the FeH/FeD index applied and related with Fe forms. I don't understand the statement at pag 11 (lines 7-13) , but since part of the discussion was based on this index within studied soils I suggest the authors to apply the already widely accepted methods proposed in international literature (Schwertmann et al., 1982; Schwertmann and Cornell, 1991.) or to explain well why they use other methodology.

We appreciate the reviewer’s desire for more clarity as it applies to the FeH/FeD index, a proxy for the crystallinity of iron, utilized in the paper. The ratio of oxalate extractable iron (FeO) to dithionite extractable Fe (FeD) is commonly utilized as a rough proxy for the crystallinity of secondary Fe-oxides within soil systems (e.g., Blume and Schwertmann, 1969; Munch and Ottow, 1980; McFadden and Hendricks, 1985; Birkeland et al., 1989; Rhoton et al., 1993, 1998; Alexander and Burt, 1996; Malucelli et al., 1999; De Jong et al., 2000; Stanjek and Marchel, 2008; Lair et al., 2009; Kabala and Zapart, 2012; Li and Richter, 2012; Aburto and Southard, 2017; Hall et al., 2018). As hydroxylamine hydrochloride was used in place of the oxalate extraction and produces similar results (e.g., Shang and Zelazny, 2008), we have substituted hydroxylamine extractable iron (FeH) for the FeO within the index. We have modified the first paragraph of the section titled “The Effect of Climate and Time on the Formation and Crystallinity of Fe-containing Amorphous Material to provide clarity in regard to the extractants used and the rough crystallinity index from lines 227 to 238–

“To examine the effect of climate and time on the formation and persistence of Fe-containing amorphous material, the abundance of Fe and Fe-associated Si within amorphous material at each field site was compared using a hydroxylamine hydrochloride dissolution (Fe_H and Si_H), which is a combined reduction and acid protonation technique selective for Fe containing amorphous material (Chester and Hughes, 1967; Chao, 1972; Chao and Zhou, 1983; Ross et al., 1985; McAlister and Smith, 1999; Shang and Zelazny, 2008). The amount of Fe within total

amorphous and crystalline (oxyhydr)oxides was examined using a citrate dithionite dissolution (Fe_D), which acts as a reducing agent for Fe^{3+} in Fe^{3+} -containing amorphous and crystalline oxides at near neutral pH conditions (Holmgren, 1967; Russell, 1979; Ericsson et al., 1984; Stucki et al., 1984; Shang and Zelazny, 2008). Together these measurements allow both a comparison of the amount of Fe in amorphous material, and the extent of crystallinity of Fe-containing phases under different climate conditions in soils of different ages. The ratio of ammonium oxalate extractable Fe to dithionite extractable Fe is commonly used as a rough crystallinity index for secondary Fe-containing material (Blume and Schwertmann, 1969; Munch and Ottow, 1980; McFadden and Hendricks, 1985; Birkeland et al., 1989; Rhoton et al., 1998; Stanjek and Marchel, 2008; Lair et al., 2009; Kabala and Zapart, 2012; Li and Richter, 2012; Aburto and Southard, 2017; Hall et al., 2018). Given the similar results produced by hydroxylamine and oxalate extractants (Ross et al., 1985; McAlister and Smith, 1999; Shang and Zelazny, 2008) we have substituted the hydroxylamine values for oxalate in the index.”

References:

- Aburto, F., and Southard, R., 2017, Refined Geomorphologic Interpretation of Glacial Deposits using Combined Soil Development Indices and LiDAR Terrain Analysis: *Soil Science Society of America Journal*, v. 81, doi:10.2136/sssaj2016.07.0211.
- Alexander, E.B., 2009, Serpentine Geoecology of the Eastern and Southeastern Margins of North America: *Northeastern Naturalist*, v. 16.
- Alexander, E.B., Adamson, C., Graham, R.C., and Zinke, P.J., 1990, Mineralogy and Classification of Soils on Serpentinized Peridotite of the Trinity Ophiolite, California: *Soil Science*, v. 149, p. 138–143.
- Alexander, E.B., and Burt, R., 1996, Soil development on moraines of Mendenhall Glacier, southeast Alaska. 1. The moraines and soil morphology: *Geoderma*, v. 72, p. 1–17, doi:10.1016/0016-7061(96)00021-3.
- Alexander, E.B., Coleman, R.G., Keeler-Wolfe, T., and Harrison, S.P., 2006, Serpentine Geoecology of Western North America: *Geology, Soils, and Vegetation*: OUP USA, 528 p.
- Alexander, E.B., Ellis, C.C., and Burke, R., 2007a, A chronosequence of soils and vegetation on serpentine terraces in the Klamath Mountains, USA: *Soil Science*, doi:10.1097/ss.0b013e31804fa22d.
- Alexander, E.B., Ellis, C.C., and Burke, R., 2007b, A chronosequence of soils and vegetation on serpentine terraces in the Klamath Mountains, USA: *Soil Science*, v. 172, doi:10.1097/ss.0b013e31804fa22d.
- Alfredsson, H., Hugelius, G., Clymans, W., Stadmark, J., Kuhry, P., and Conley, D.J., 2015, Amorphous silica pools in permafrost soils of the Central Canadian Arctic and the potential impact of climate change: *Biogeochemistry*, v. 124, doi:10.1007/s10533-015-0108-1.
- Anthony, J.W., Bideaux, R.A., Bladh, K.W., and Nichols, M.C., 2003, *Handbook of Mineralogy Crystal Data*: Mineralogical Society of America,.

- Arguez, A., Durre, I., Applequist, S., Vose, R.S., Squires, M.F., Yin, X., Heim, R.R., and Owen, T.W., 2012, NOAA's 1981-2010 U.S. climate normals: *Bulletin of the American Meteorological Society*, v. 93, p. 1687–1697, doi:10.1175/BAMS-D-11-00197.1.
- Bedford, C.C., Bridges, J.C., Schwenzler, S.P., Wiens, R.C., Rampe, E.B., Frydenvang, J., and Gasda, P.J., 2019, Alteration trends and geochemical source region characteristics preserved in the fluviolacustrine sedimentary record of Gale crater, Mars: *Geochimica et Cosmochimica Acta*, v. 246, p. 234–266, doi:10.1016/j.gca.2018.11.031.
- Birkeland, P.W., Burke, R.M., and Benedict, J.B., 1989, Pedogenic gradients for iron and aluminum accumulation and phosphorus depletion in arctic and alpine soils as a function of time and climate: *Quaternary Research*, v. 32, p. 193–204, doi:10.1016/0033-5894(89)90075-6.
- Bishop, J.L., Fairén, A.G., Michalski, J.R., Gago-Duport, L., Baker, L.L., Velbel, M.A., Gross, C., and Rampe, E.B., 2018, Surface clay formation during short-term warmer and wetter conditions on a largely cold ancient Mars: *Nature Astronomy*, v. 2, p. 206–213, doi:10.1038/s41550-017-0377-9.
- Bishop, J.L., and Rampe, E.B., 2016, Evidence for a changing Martian climate from the mineralogy at Mawrth Vallis: *Earth and Planetary Science Letters*, v. 448, p. 42–48, doi:10.1016/j.epsl.2016.04.031.
- Blake, D. et al., 2012, Characterization and calibration of the CheMin mineralogical instrument on Mars Science Laboratory: *Space Science Reviews*, v. 170, p. 341–399, doi:10.1007/s11214-012-9905-1.
- Blume, H.P., and Schwertmann, U., 1969, Genetic Evaluation of Profile Distribution of Aluminum, Iron, and Manganese Oxides: *Soil Science Society of America Journal*, v. 33, doi:10.2136/sssaj1969.03615995003300030030x.
- Bonifacio, E., Zanini, E., and Boero, V., 1997, Pedogenesis in a soil catena on serpentinite in north-western Italy: *Geoderma*, v. 75, p. 33–51.
- Bouchard, A., Hay, S., Gauvin, C., and Bergeron, Y., 1986, Rare Vascular Plants of Gros Morne National Park, Newfoundland, Canada: *Rhodora*, v. 88, p. 481–502.
- Brouillet, L., Hay, S., Turcotte, P., Bouchard, A., and Marie-Victorin, H., 1998, The alpine vascular flora of the Big Level Plateau, Gros Morne National Park, Newfoundland.: *Geographie Physique et Quaternaire*, v. 52, p. 175–193, doi:10.7202/004774ar.
- Butt, C.R.M., and Cluzel, D., 2013, Nickel laterite ore deposits: Weathered serpentinites: *Elements*, v. 9, doi:10.2113/gselements.9.2.123.
- Caillaud, J., Proust, D., Righi, D., and Martin, F., 2004, Fe-rich clays in a weathering profile developed from serpentinite: *Clays and Clay Minerals*, v. 52, p. 779–791, doi:10.1346/CCMN.2004.05206013.
- Chao, T.T., 1972, Selective Dissolution of Manganese Oxides from Soils and Sediments with Acidified Hydroxylamine Hydrochloride: *Soil Science Society of America Journal*, v. 36, p. 764–768.
- Chao, T.T., and Zhou, L., 1983, Extraction Techniques for Selective Dissolution of Amorphous Iron Oxides from Soils and Sediments: *Soil Science Society of America Journal*, v. 47, p. 225–232, doi:10.2136/sssaj1983.03615995004700020010x.

- Chester, R., and Hughes, M.J., 1967, A chemical technique for the separation of ferro-manganese minerals, carbonate minerals and adsorbed trace elements from pelagic sediments: *Chemical Geology*, v. 2, p. 249–262.
- Chevrier, V., and Mathé, P.E., 2007, Mineralogy and evolution of the surface of Mars: A review: *Planetary and Space Science*, v. 55, p. 289–314, doi:10.1016/j.pss.2006.05.039.
- Childs, C.W., 1992, Ferrihydrite: A review of structure, properties and occurrence in relation to soils: *Zeitschrift für Pflanzenernährung und Bodenkunde*, v. 155, doi:10.1002/jpln.19921550515.
- Chipera, S.J. et al., 2023, Mineralogical Investigation of Mg-Sulfate at the Canaima Drill Site, Gale Crater, Mars: *Journal of Geophysical Research: Planets*, v. 128, doi:10.1029/2023JE008041.
- Chipera, S.J., and Bish, D.L., 2002, FULLPAT: A full-pattern quantitative analysis program for X-ray powder diffraction using measured and calculated patterns: *Journal of Applied Crystallography*, v. 35, p. 744–749, doi:10.1107/S0021889802017405.
- Cismasu, A.C., Michel, F.M., Tcaciuc, A.P., Tyliszczak, T., and Brown, Jr, G.E., 2011, Composition and structural aspects of naturally occurring ferrihydrite: *Comptes Rendus Geoscience*, v. 343, doi:10.1016/j.crte.2010.11.001.
- Cudennec, Y., and Lecerf, A., 2006, The transformation of ferrihydrite into goethite or hematite, revisited: *Journal of Solid State Chemistry*, doi:10.1016/j.jssc.2005.11.030.
- D’Amico, M.E., Freppaz, M., Leonelli, G., Bonifacio, E., and Zanini, E., 2015, Early stages of soil development on serpentinite: the proglacial area of the Verra Grande Glacier, Western Italian Alps: *Journal of Soils and Sediments*, doi:10.1007/s11368-014-0893-5.
- Dietze, M., Bartel, S., Lindner, M., and Kleber, A., 2012, Formation mechanisms and control factors of vesicular soil structure: *Catena*, v. 99, doi:10.1016/j.catena.2012.06.011.
- Drees, L.R., Wilding, L.P., Smeck, N.E., and Senkayi, A.L., 2018, Silica in soils: Quartz and disordered silica polymorphs, *in* *Minerals in Soil Environments*, doi:10.2136/sssabookser1.2ed.c19.
- Drits, V.A., Sakharov, B.A., Salyn, A.L., and Manceau, A., 1993, Structural Model for Ferrihydrite: *Clay Minerals*, v. 28, doi:10.1180/claymin.1993.028.2.02.
- Du, P. et al., 2020, Effects of Environmental Fe Concentrations on Formation and Evolution of Allophane in Al-Si-Fe Systems: Implications for Both Earth and Mars: *Journal of Geophysical Research: Planets*, v. 125, doi:10.1029/2020JE006590.
- Eggleton, R.A., and Fitzpatrick, R.W., 1988, New Data and a Revised Structural Model for Ferrihydrite: *Clays and Clay Minerals*, v. 36, doi:10.1346/CCMN.1988.0360203.
- Eggleton, R.A., and Tilley, D.B., 1998, Hisingerite: A ferric kaolin mineral with curved morphology: *Clays and Clay Minerals*, v. 46, p. 400–413, doi:10.1346/CCMN.1998.0460404.
- Environment Canada, 2010, Canadian Climate Normals 1981-2010 Station Data. Cow Head. Climate ID: 8401335; http://climate.weather.gc.ca/climate_normals/index_e.html#1981 (accessed October 2023).

- Ericsson, T., Linares, J., and Lotse, E., 1984, A Mossbauer Study of the Effect of Dithionite/Citrate/Bicarbonate Treatment on a Vermiculite, A Smectite, and a Soil: *Clay Minerals*, v. 19, p. 85–91.
- Farmer, V.C., 1992, Possible Confusion Between So-Called Ferrihydrites and Hisingerites: *Clay Minerals*, v. 27, p. 373–378, doi:10.1180/claymin.1992.027.3.10.
- Feldman, A., Hausrath, E., Rampe, E., Sharp, T., Tschauer, O., Lanzirrotti, A., and Newville, M., 2023, Nanoscale Analyses of X-ray Amorphous Material from Terrestrial Ultramafic Soils Record Signatures of Environmental Conditions Useful for Interpreting Past Martian Conditions: *Geochemistry, Geophysics, Geosystems (Under Review)*,.
- Garwood, J.M., and Welsh, H.H., 2007, Ecology of the Cascades Frog (*Rana cascadae*) and Interactions with Garter Snakes and Nonnative Trout in the Trinity Alps Wilderness, California.:
- Gaudin, A., Dehouck, E., and Mangold, N., 2011, Evidence for weathering on early Mars from a comparison with terrestrial weathering profiles: *Icarus*, v. 216, p. 257–268, doi:10.1016/j.icarus.2011.09.004.
- Gleeson, S.A., Herrington, R.J., Durango, J., Velásquez, C.A., and Koll, G., 2004, The mineralogy and geochemistry of the Cerro Matoso S.A. Ni Laterite deposit, Montelíbano, Colombia: *Economic Geology*, v. 99, doi:10.2113/gsecongeo.99.6.1197.
- Goudge, T.A., Russell, J.M., Mustard, J.F., Head, J.W., and Bijaksana, S., 2017, A 40,000 yr record of clay mineralogy at Lake Towuti, Indonesia: Paleoclimate reconstruction from reflectance spectroscopy and perspectives on paleolakes on Mars: *Bulletin of the Geological Society of America*, v. 129, doi:10.1130/B31569.1.
- Grosch, E.G., Bishop, J.L., Mielke, C., Maturilli, A., and Helbert, J., 2021, Early Archean alteration minerals in mafic-ultramafic rocks of the Barberton greenstone belt as petrological analogs for clay mineralogy on Mars: *American Mineralogist*, v. 106, doi:10.2138/am-2021-7656.
- Gutiérrez-Castorena, M.D.C., Stoops, G., Ortiz Solorio, C.A., and López Avila, G., 2005, Amorphous silica materials in soils and sediments of the Ex-Lago de Texcoco, Mexico: An explanation for its subsidence: *Catena*, v. 60, doi:10.1016/j.catena.2004.11.005.
- Hall, S.J., Berhe, A.A., and Thompson, A., 2018, Order from disorder: do soil organic matter composition and turnover co-vary with iron phase crystallinity? *Biogeochemistry*, v. 140, doi:10.1007/s10533-018-0476-4.
- Heiri, O., Lotter, A.F., and Lemcke, G., 2001, Loss on ignition as a method for estimating organic and carbonate content in sediments: reproducibility and comparability of results Oliver: *Journal of Paleolimnology*, v. 25, p. 101–110, doi:10.1016/0009-2541(93)90140-E.
- Holder, C.F., and Schaak, R.E., 2019, Tutorial on Powder X-ray Diffraction for Characterizing Nanoscale Materials: *ACS Nano*, v. 13, p. 7359–7365, doi:10.1021/acsnano.9b05157.
- Holmgren, G.G.S., 1967, A Rapid Citrate-Dithionite Extractable Iron Procedure: *Soil Science Society of America Journal*, v. 31, p. 210–211, doi:10.2136/sssaj1967.03615995003100020020x.

- Horgan, B., and Bell, J.F., 2012, Widespread weathered glass on the surface of Mars: *Geology*, v. 40, p. 391–394, doi:10.1130/G32755.1.
- Istok, J.D.D., and Harward, M.E.E., 1982, Influence of Soil Moisture on Smectite Formation in Soils Derived from Serpentinite: *Soil Science Society of America Journal*, v. 46, p. 1106–1108.
- Jansen, E., Kyek, A., Schäfer, W., and Schwertmann, U., 2002, The structure of six-line ferrihydrite: *Applied Physics A: Materials Science and Processing*, v. 74, doi:10.1007/s003390101175.
- Jenny, H., 1941, *Factors of Soil Formation*: New York, 281 p.
- Jiang, Z., Liu, Q., Roberts, A.P., Barrón, V., Torrent, J., and Zhang, Q., 2018, A new model for transformation of ferrihydrite to hematite in soils and sediments: *Geology*, v. 46, doi:10.1130/G45386.1.
- De Jong, E., Pennock, D.J., and Nestor, P.A., 2000, Magnetic susceptibility of soils in different slope positions in Saskatchewan, Canada: *Catena*, v. 40, doi:10.1016/S0341-8162(00)00080-1.
- Kabala, C., and Zapart, J., 2012, Initial soil development and carbon accumulation on moraines of the rapidly retreating Werenskiöld Glacier, SW Spitsbergen, Svalbard archipelago: *Geoderma*, doi:10.1016/j.geoderma.2012.01.025.
- Kiczka, M., Wiederhold, J.G., Frommer, J., Voegelin, A., Kraemer, S.M., Bourdon, B., and Kretzschmar, R., 2011, Iron speciation and isotope fractionation during silicate weathering and soil formation in an alpine glacier forefield chronosequence: *Geochimica et Cosmochimica Acta*, v. 75, doi:10.1016/j.gca.2011.07.008.
- Klopprogge, J.T., Komarneni, S., and Amonette, J.E., 1999, Synthesis of smectite clay minerals: A critical review: *Clays and Clay Minerals*, v. 47, p. 529–554, doi:10.1346/CCMN.1999.0470501.
- Klug, H.P., and Alexander, L.E., 1974, *X-ray Diffraction Procedures: for Polycrystalline and Amorphous Materials*: Wiley, 992 p.
- Kohyama, N., and Sudo, T., 1975, Hisingerite occurring as a weathering product of iron-rich saponite: *Clays and Clay Minerals*, v. 23, p. 215–218, doi:10.1346/CCMN.1975.0230309.
- Lair, G.J., Zehetner, F., Hrachowitz, M., Franz, N., Maringer, F.J., and Gerzabek, M.H., 2009, Dating of soil layers in a young floodplain using iron oxide crystallinity: *Quaternary Geochronology*, v. 4, doi:10.1016/j.quageo.2008.11.003.
- Lee, B.D., Sears, S.K., Graham, R.C., Amrhein, C., and Vali, H., 2003, Secondary Mineral Genesis from Chlorite and Serpentine in an Ultramafic Soil Toposequence: *Soil Science Society of America Journal*, v. 67, p. 1309–1317.
- Leshin, L.A. et al., 2013, Volatile, isotope, and organic analysis of martian fines with the Mars curiosity rover: *Science*, v. 341, doi:10.1126/science.1238937.
- Li, J., and Richter, D.D., 2012, Effects of two-century land use changes on soil iron crystallinity and accumulation in Southeastern Piedmont region, USA: *Geoderma*, v. 173–174, doi:10.1016/j.geoderma.2011.12.021.

- Malucelli, F., Terribile, F., and Colombo, C., 1999, Mineralogy, micromorphology and chemical analysis of andosols on the Island of Sao Miguel (Azores): *Geoderma*, v. 88, doi:10.1016/S0016-7061(98)00081-0.
- Manceau, A., 2009, Evaluation of the structural model for ferrihydrite derived from real-space modelling of high-energy X-ray diffraction data: *Clay Minerals*, v. 44, doi:10.1180/claymin.2009.044.1.19.
- Mayhew, L.E., and Ellison, E.T., 2020, A synthesis and meta-analysis of the Fe chemistry of serpentinites and serpentine minerals: *Philosophical Transactions of the Royal Society A: Mathematical, Physical and Engineering Sciences*, v. 378, doi:10.1098/rsta.2018.0420.
- McAlister, J.J., and Smith, B.J., 1999, Selectivity of ammonium acetate, hydroxylamine hydrochloride, and oxalate/ascorbic acid solutions for the speciation of Fe, Mn, Zn, Cu, Ni, and Al in early tertiary paleosols: *Microchemical Journal*, v. 63, p. 415–426, doi:10.1006/mchj.1999.1798.
- McFadden, L.D., and Hendricks, D.M., 1985, Changes in the content and composition of pedogenic iron oxyhydroxides in a chronosequence of soils in southern California: *Quaternary Research*, v. 23, p. 189–204, doi:10.1016/0033-5894(85)90028-6.
- McKeague, J.A., 1967, An Evaluation of 0.1 M Pyrophosphate and Pyrophosphate-Dithionite in Comparison With Oxalate as Extractants of the Accumulation Products in Podzols and Some Other Soils: *Canadian Journal of Soil Science*, v. 47, p. 95–99.
- McLennan, S.M., Grotzinger, J.P., Hurowitz, J.A., and Tosca, N.J., 2019, The sedimentary cycle on early mars: *Annual Review of Earth and Planetary Sciences*, v. 47, doi:10.1146/annurev-earth-053018-060332.
- Michel, F.M., Ehm, L., Antao, S.M., Lee, P.L., Chupas, P.J., Liu, G., Strongin, D.R., Schoonen, M.A.A., Phillips, B.L., and Parise, J.B., 2007, The structure of ferrihydrite, a nanocrystalline material: *Science*, v. 316, doi:10.1126/science.1142525.
- Milliken, R.E. et al., 2008, Opaline silica in young deposits on Mars: *Geology*, v. 36, doi:10.1130/G24967A.1.
- Mohr, J.A., Whitlock, C., and Skinner, C.N., 2000, Postglacial vegetation and fire history, eastern Klamath Mountains, California, USA: *Holocene*, v. 10, p. 587–601, doi:10.1191/095968300675837671.
- Morris, R. V et al., 2013, The Amorphous Component in Martian Basaltic Soil in Global Perspective From MSL and MER Missions.:
- Mosser-Ruck, R., Cathelineau, M., Guillaume, D., Charpentier, D., Rousset, D., Barres, O., and Michau, N., 2010, Effects of temperature, pH, and iron/clay and liquid/clay ratios on experimental conversion of dioctahedral smectite to berthierine, chlorite, vermiculite, or saponite: *Clays and Clay Minerals*, v. 58, p. 280–291, doi:10.1346/CCMN.2010.0580212.
- Munch, J.C., and Ottow, J.C., 1980, Preferential reduction of amorphous to crystalline iron oxides by bacterial activity: *Soil Science*, v. 129, doi:10.1097/00010694-198001000-00004.

- Pan, L., and Ehlmann, B.L., 2014, Phyllosilicate and hydrated silica detections in the knobby terrains of Acidalia Planitia, northern plains, Mars: *Geophysical Research Letters*, v. 41, doi:10.1002/2014GL059423.
- Parfitt, R.L., Childs, C.W., and Eden, D.N., 1988, Ferrihydrite and allophane in four Andepts from Hawaii and implications for their classification: *Geoderma*, v. 41, p. 223–241, doi:10.1016/0016-7061(88)90062-6.
- Parfitt, R.L., and Kimble, J.M., 1989, Conditions for Formation of Allophane in Soils: *Soil Science Society of America Journal*, v. 53, doi:10.2136/sssaj1989.03615995005300030057x.
- Petit, S., Baron, F., and Decarreau, A., 2017, Synthesis of nontronite and other Fe-rich smectites: a critical review: *Clay Minerals*, v. 52, p. 469–483, doi:10.1180/claymin.2017.052.4.05.
- Poulet, F., Mangold, N., Loizeau, D., Bibring, J., Langevin, Y., Michalski, J., and Gondet, B., 2008, Abundance of minerals in the phyllosilicate-rich units on Mars: *Astronomy and Astrophysics*, v. 44, p. 41–45, doi:10.1051/0004-6361.
- Rampe, E.B. et al., 2020a, Mineralogy and geochemistry of sedimentary rocks and eolian sediments in Gale crater, Mars: A review after six Earth years of exploration with Curiosity: *Geochemistry*, v. 80, p. 125605, doi:10.1016/j.chemer.2020.125605.
- Rampe, E.B. et al., 2020b, Mineralogy of Vera Rubin Ridge From the Mars Science Laboratory CheMin Instrument: *Journal of Geophysical Research: Planets*, v. 125, p. 1–31, doi:10.1029/2019JE006306.
- Rampe, E.B., Horgan, B.H.N., Smith, R.J., Scudder, N.A., Bamber, E.R., Rutledge, A.M., and Christoffersen, R., 2022, A mineralogical study of glacial flour from Three Sisters, Oregon: An analog for a cold and icy early Mars: *Earth and Planetary Science Letters*, v. 584, doi:10.1016/j.epsl.2022.117471.
- Rampe, E.B., Kraft, M.D., Sharp, T.G., Golden, D.C., Ming, D.W., and Christensen, P.R., 2012a, Allophane detection on Mars with Thermal Emission spectrometer data and implications for regional-scale chemical weathering processes: *Geology*, v. 40, p. 995–998, doi:10.1130/G33215.1.
- Rampe, E.B., Kraft, M.D., Sharp, T.G., Golden, D.C., Ming, D.W., and Christensen, P.R., 2012b, Allophane detection on Mars with Thermal Emission spectrometer data and implications for regional-scale chemical weathering processes: *Geology*, v. 40, p. 995–998, doi:10.1130/G33215.1.
- Rancourt, D.G., and Meunier, J.F., 2008, Constraints on structural models of ferrihydrite as a nanocrystalline material: *American Mineralogist*, v. 93, doi:10.2138/am.2008.2782.
- Rapin, W. et al., 2018, In Situ Analysis of Opal in Gale Crater, Mars: *Journal of Geophysical Research: Planets*, v. 123, doi:10.1029/2017JE005483.
- Rhoton, F.E., Bigham, J.M., and Schulze, D.G., 1993, Properties of Iron—Manganese Nodules from a Sequence of Eroded Fragipan Soils: *Soil Science Society of America Journal*, v. 57, doi:10.2136/sssaj1993.03615995005700050037x.
- Rhoton, F.E., Römken, M.J.M., and Lindbo, D.L., 1998, Iron Oxides Erodibility Interactions for Soils of the Memphis Catena: *Soil Science Society of America Journal*, v. 62, doi:10.2136/sssaj1998.03615995006200060030x.

- Roberts, B.A., 1980, Some Chemical and Physical Properties of Serpentine Soils From Western Newfoundland: *Canadian Journal of Soil Science*, v. 60, p. 231–240.
- Ross, G.J., Wang, C., and Schuppli, P.A., 1985, Hydroxylamine and Ammonium Oxalate Solutions As Extractants For Iron and Aluminum From Soils: *Soil Science Society of America Journal*, v. 49, p. 783–785.
- Ruff, S.W. et al., 2011, Characteristics, distribution, origin, and significance of opaline silica observed by the Spirit rover in Gusev crater, Mars: *Journal of Geophysical Research: Planets*, v. 116, doi:10.1029/2010JE003767.
- Ruff, S.W., and Hamilton, V.E., 2017, Wishstone to Watchtower: Amorphous alteration of plagioclase-rich rocks in Gusev crater, Mars: *American Mineralogist*, v. 102, p. 235–251, doi:10.2138/am-2017-5618.
- Russell, J.D., 1979, Infrared and Mössbauer Studies of Reduced Nontronites: *Clays and Clay Minerals*, v. 27, p. 63–71, doi:10.1346/ccmn.1979.0270108.
- De Saussure, H.B., 1796, *Voyages dans les Alpes*: v. 3.
- Schwertmann, U., 1993, Relations between iron oxides, soil color, and soil formation: *Soil color. Proc. symposium, San Antonio, 1990*,.
- Schwertmann, U., Friedl, J., and Stanjek, H., 1999, From Fe(III) ions to ferrihydrite and then to hematite: *Journal of Colloid and Interface Science*, v. 209, doi:10.1006/jcis.1998.5899.
- Schwertmann, U., and Murad, E., 1983, Effect of pH on the formation of goethite and hematite from ferrihydrite.: *Clays & Clay Minerals*, v. 31, doi:10.1346/CCMN.1983.0310405.
- Shang, C., and Zelazny, L.W., 2008, Selective Dissolution Techniques for Mineral Analysis of Soils and Sediments, in ULERY, A.L. and RICHARD DREES, L. eds., *Methods of Soil Analysis Part 5 - Mineralogical Methods*, Soil Science Society of America, p. 33–80, doi:10.2136/sssabookser5.5.c3.
- Sheppard, R.Y., Milliken, R.E., Russell, J.M., Dyar, M.D., Sklute, E.C., Vogel, H., Melles, M., Bijaksana, S., Morlock, M.A., and Hasberg, A.K.M., 2019, Characterization of Iron in Lake Towuti sediment: *Chemical Geology*, v. 512, doi:10.1016/j.chemgeo.2019.02.029.
- Siebach, K.L., Baker, M.B., Grotzinger, J.P., Mclennan, S.M., Gellert, R., Thompson, L.M., and Hurowitz, J.A., 2017, Sorting out compositional trends in sedimentary rocks of the Bradbury group (Aeolis Palus), Gale crater, Mars: *Journal of Geophysical Research: Planets*, v. 122, p. 295–328, doi:10.1038/175238c0.
- Skinner, C.N., Taylor, A.H., and Agee, J.K., 2006, Klamath Mountains bioregion: Fire in California's Ecosystems, p. 170–194.
- Smith, R.J., and Horgan, B.H.N., 2021, Nanoscale Variations in Natural Amorphous and Nanocrystalline Weathering Products in Mafic to Intermediate Volcanic Terrains on Earth: Implications for Amorphous Detections on Mars: *Journal of Geophysical Research: Planets*, v. 126, p. 1–30, doi:10.1029/2020JE006769.

- Smith, R.J., McLennan, S.M., Achilles, C.N., Dehouck, E., Horgan, B.H.N., Mangold, N., Rampe, E.B., Salvatore, M., Siebach, K.L., and Sun, V., 2021, X-Ray Amorphous Components in Sedimentary Rocks of Gale Crater, Mars: Evidence for Ancient Formation and Long-Lived Aqueous Activity: *Journal of Geophysical Research: Planets*, v. 126, doi:10.1029/2020JE006782.
- Smith, R.J., Rampe, E.B., Horgan, B.H.N., and Dehouck, E., 2018, Deriving amorphous component abundance and composition of rocks and sediments on Earth and Mars: *Journal of Geophysical Research: Planets*, v. 123, p. 2485–2505, doi:10.1029/2018JE005612.
- Soil Survey Staff, 2004, *Soil Survey Laboratory Methods Manual: Soil Survey Investigations Report*, v. 42, p. 700.
- Stanjek, H., and Marchel, C., 2008, Linking the redox cycles of Fe oxides and Fe-rich clay minerals: an example from a palaeosol of the Upper Freshwater Molasse: *Clay Minerals*, v. 43, doi:10.1180/claymin.2008.043.1.05.
- Stucki, J.W., Golden, D.C., and Roth, C.B., 1984, Preparation and Handling of Dithionite-Reduced Smectite Suspensions.: *Clays and Clay Minerals*, v. 32, p. 191–197, doi:10.1346/CCMN.1984.0320306.
- Stucki, J.W., Goodman, B.A., and Schwertmann, U., 1988, *Iron in Soils and Clay Minerals*: 893 p., [https://books.google.se/books?hl=en&lr=&id=32roCAAQBAJ&oi=fnd&pg=PR19&dq=Iron+in+Soils+and+Clay+Minerals&ots=-IZMmvuAPH&sig=YcGhnOHF3tdaA388SD3_rHHlt7Y&redir_esc=y#v=onepage&q=Iron in Soils and Clay Minerals&f=false](https://books.google.se/books?hl=en&lr=&id=32roCAAQBAJ&oi=fnd&pg=PR19&dq=Iron+in+Soils+and+Clay+Minerals&ots=-IZMmvuAPH&sig=YcGhnOHF3tdaA388SD3_rHHlt7Y&redir_esc=y#v=onepage&q=Iron+in+Soils+and+Clay+Minerals&f=false).
- Sun, V.Z., and Milliken, R.E., 2018, Distinct Geologic Settings of Opal-A and More Crystalline Hydrated Silica on Mars: *Geophysical Research Letters*, v. 45, doi:10.1029/2018GL078494.
- Sutter, B. et al., 2017, Evolved gas analyses of sedimentary rocks and eolian sediment in Gale Crater, Mars: Results of the Curiosity rover's sample analysis at Mars instrument from Yellowknife Bay to the Namib Dune: *Journal of Geophysical Research: Planets*, v. 122, p. 2574–2609, doi:10.1002/2016JE005225.
- Thorpe, M.T. et al., 2022, Mars Science Laboratory CheMin Data From the Glen Torridon Region and the Significance of Lake-Groundwater Interactions in Interpreting Mineralogy and Sedimentary History: *Journal of Geophysical Research: Planets*, v. 127, p. e2021JE007099, doi:10.1029/2021JE007099.
- Torrent, J., Schwertmann, U., Fechter, H., and Alferez, F., 1983, Quantitative relationships between soil color and hematite content: *Soil Science*, v. 136, doi:10.1097/00010694-198312000-00004.
- Turk, J.K., and Graham, R.C., 2011, Distribution and Properties of Vesicular Horizons in the Western United States: *Soil Science Society of America Journal*, v. 75, p. 1449–1461, doi:10.2136/sssaj2010.0445.
- Tutolo, B.M., Evans, B.W., and Kuehner, S.M., 2019, Serpentine–Hisingerite solid solution in altered ferroan peridotite and Olivine Gabbro: *Minerals*, v. 9, p. 1–14, doi:10.3390/min9010047.
- Tutolo, B.M., and Tosca, N.J., 2023, Observational constraints on the process and products of Martian serpentinization: *Science Advances*, v. 9, doi:10.1126/sciadv.add8472.

- Wada, K., Henmi, T., Yoshinaga, N., and Patterson, S.H., 1972, Imogolite and allophane formed in saprolite of basalt on Maui, Hawaii: *Clays and Clay Minerals*, v. 20, p. 375–380, doi:10.1346/ccmn.1972.0200605.
- Weitz, C.M., and Bishop, J.L., 2019, Formation of Clays, Ferrihydrite, and Possible Salts in Hydrae Chasma, Mars: *Icarus*, v. 319, doi:10.1016/j.icarus.2018.09.007.
- Weitz, C.M., Bishop, J.L., Baker, L.L., and Berman, D.C., 2014, Fresh exposures of hydrous Fe-bearing amorphous silicates on Mars: *Geophysical Research Letters*, v. 41, p. 8744–8751, doi:10.1002/2014GL062065.
- Weitz, C.M., Bishop, J.L., and Grant, J.A., 2013, Gypsum, opal, and fluvial channels within a trough of noctis labyrinthus, mars: Implications for aqueous activity during the late hesperian to amazonian: *Planetary and Space Science*, v. 87, doi:10.1016/j.pss.2013.08.007.
- West, N.E., 1983, Great Basin- Colorado Plateau sagebrush semi-desert (*Artemisia*): Temperate deserts and semi-deserts,.
- Wildman, W.E., Whittig, L.D., and Jackson, M.L., 1971, Serpentine stability in relation to formation of iron-rich montmorillonite in some California soils: *Amer. Mineral*, v. 56, p. 587–602.
- Ziegler, K., Hsieh, J.C.C., Chadwick, O.A., Kelly, E.F., Hendricks, D.M., and Savine, S.M., 2003, Halloysite as a kinetically controlled end product of arid-zone basalt weathering: *Chemical Geology*, v. 202, p. 461–478, doi:10.1016/j.chemgeo.2002.06.001.

8th May 24

Dear Dr Feldman,

Please allow us to apologise for the long delay in ending a decision on your manuscript titled "Fe-rich X-Ray Amorphous Material Records Past Climate and Persistence of Water on Mars". It has now been seen by our reviewers, whose comments appear below. In light of their advice we are delighted to say that we are happy, in principle, to publish a suitably revised version in Communications Earth & Environment under the open access CC BY license (Creative Commons Attribution v4.0 International License).

We therefore invite you to edit your manuscript to comply with our format requirements and to maximise the accessibility and therefore the impact of your work.

EDITORIAL REQUESTS:

*****Please take care to match our formatting and policy requirements. We will check revised manuscript and return manuscripts that do not comply. Such requests will lead to delays. *****

SUBMISSION INFORMATION:

OPEN ACCESS:

Communications Earth & Environment is a fully open access journal. Articles are made freely accessible on publication under a CC BY license (Creative Commons Attribution 4.0 International License). This license allows maximum dissemination and re-use of open access materials and is preferred by many research funding bodies.

For further information about article processing charges, open access funding, and advice and support from Nature Research, please visit <https://www.nature.com/commsenv/article-processing-charges>

At acceptance, you will be provided with instructions for completing this CC BY license on behalf of all authors. This grants us the necessary permissions to publish your paper. Additionally, you will be

asked to declare that all required third party permissions have been obtained, and to provide billing information in order to pay the article-processing charge (APC).

[link redacted]

Best regards,

Mojtaba Fakhraee, PhD
Editorial Board Member
Communications Earth & Environment
orcid.org/0000-0002-2461-6374

Joe Aslin
Deputy Editor,
Communications Earth & Environment
<https://www.nature.com/commsenv/>
Twitter: @CommsEarth

REVIEWERS' COMMENTS:

Reviewer #1 (Remarks to the Author):

All of my comments have been well addressed and this manuscript has been improved a lot in quality after the revision. Therefore, I am happy to recommend it to be published ASAP.

Reviewer #3 (Remarks to the Author):

Second revision of the paper “Fe-rich X-Ray Amorphous Material Records Past Climate and Persistence of Water on Mars”, I am satisfied with the corrections and enhancements made in response to my comments. In light of these improvements, I recommend the publication of the revised manuscript in its current form. I thank the authors for helping to understand their work and the implications on the future study of Mars soils.

Furthermore, I would like to suggest for authors future research, that a similar research approach be applied Differential XRD technology, coupled with an appropriate correction factor for total elements and NIR spectroscopy analysis, I believe that pursuing this direction could lead to groundbreaking discoveries and contribute significantly to the field as it would allow to understand the presence of Fe-rich amorphous material at Gale crater and elsewhere on Mars. Congratulation

for the interesting paper.

Reviewer #4 (Remarks to the Author):

I found the work very interesting with a plethora of data related to soil mineralogy. I believe that the main challenge to compare soils on Earth with analogue data on Mars has been accomplished to a major part. More of my queries have been cleared by the responses to the other reviewer comments, so my main concern focus to the lack of analysis of a soil material with a great resemblance with Martian surface materials.

The soil Fe-Mn concretions and nodules represent morphological features with extreme heterogeneity in relation to the surrounding soil reflecting different pedogenetic conditions. Many studies have show that the concretions and nodules contain skeletal grains of primary minerals (like quartz, feldspar etc.), pores cemented together with a brown or dark matrix which is consisted of Fe-Mn oxides, hydroxides and oxyhydroxides and clay minerals

It will be a fruitfull addition of relevant information of these unique redoximorphic soil features that have pedogenic origin reflecting different pedoclimatic conditions as a future potential analogue to Martian surface materials (<https://doi.org/10.1016/j.catena.2019.104106>;
<https://doi.org/10.1016/j.geoderma.2013.11.008>;
<https://doi.org/10.1016/j.geoderma.2021.115445>; <https://doi.org/10.1080/0365034042000216149>;
<https://doi.org/10.1016/j.jsames.2023.104424>)

01-03-2023

Dear Professor Fakhraee and Editorial Staff,

Thank you very much for the reviews of our manuscript, and for the opportunity to revise, format, and resubmit the manuscript. We have formatted the manuscript in response to the recommendations and requirements.

Please find below our responses to the reviewers' comments outside of our formatting changes. The comments from the reviewer are indicated in italics and our response is given in plain text, Please feel free to contact me if you have any questions and thank you again.

Sincerely,

Anthony Feldman

Reviewer(s)' Comments to Author:

Reviewer #1 (Remarks to the Author):

All of my comments have been well addressed and this manuscript has been improved a lot in quality after the revision. Therefore, I am happy to recommend it to be published ASAP.

We appreciate the opportunity to have revised the paper to the reviewers satisfaction.

Reviewer #3 (Remarks to the Author):

Second revision of the paper "Fe-rich X-Ray Amorphous Material Records Past Climate and Persistence of Water on Mars", I am satisfied with the corrections and enhancements made in response to my comments. In light of these improvements, I recommend the publication of the revised manuscript in its current form. I thank the authors for helping to understand their work and the implications on the future study of Mars soils.

Furthermore, I would like to suggest for authors future research, that a similar research approach be applied Differential XRD technology, coupled with an appropriate correction factor for total elements and NIR spectroscopy analysis, I believe that pursuing this direction could lead to groundbreaking discoveries and contribute significantly to the field as it would allow to

understand the presence of Fe-rich amorphous material at Gale crater and elsewhere on Mars. Congratulation for the interesting paper.

We likewise appreciate the suggestions made by reviewer 3 and believe the revisions made based on their suggestions have improved the publication. We also appreciate and intend to follow up on their suggestions for future research directions that could build upon this current work.

Reviewer #4 (Remarks to the Author):

I found the work very interesting with a plethora of data related to soil mineralogy. I believe that the main challenge to compare soils on Earth with analogue data on Mars has been accomplished to a major part. More of my queries have been cleared by the responses to the other reviewer comments, so my main concern focus to the lack of analysis of a soil material with a great resemblance with Martian surface materials.

The soil Fe-Mn concretions and nodules represent morphological features with extreme heterogeneity in relation to the surrounding soil reflecting different pedogenetic conditions. Many studies have show that the concretions and nodules contain skeletal grains of primary minerals (like quartz, feldspar etc.), pores cemented together with a brown or dark matrix which is consisted of Fe-Mn oxides, hydroxides and oxyhydroxides and clay minerals. It will be a fruitfull addition of relevant information of these unique redoximorphic soil features that have pedogenic origin reflecting different pedoclimatic conditions as a future potential analogue to Martian surface materials

(<https://doi.org/10.1016/j.catena.2019.104106>; <https://doi.org/10.1016/j.geoderma.2013.11.008>; <https://doi.org/10.1016/j.geoderma.2021.115445>; <https://doi.org/10.1080/0365034042000216149>; <https://doi.org/10.1016/j.jsames.2023.104424>)

We appreciate the reviewer noting the complexity of the soil mineralogical systems we have investigated and the note that our central aim to examine climatic effects on amorphous material formation has been accomplished satisfactorily.

We also greatly appreciate the author's interest in soil concretions, particularly in regards to the formation of mineralogically and chemically complex nodules relevant to Martian weathering questions within heterogenous soil environments. We concur that future work should examine the effects of varying environmental conditions on the formation of such nodules that combine primary minerals and a plethora of secondary phases including oxides, oxyhydroxides, clay minerals, and likely amorphous gels.

As stated in the prior response to reviewers, the focus of this particular manuscript is on bulk-scale variations in the crystallinity of secondary Fe containing material and its applicability to Martian mineral assemblages. We thus believe that while future work should investigate Fe-Mn

oxides and the formation of concretions within terrestrial soils that bear chemical relevance to martian materials, that such work falls outside the scope of this particular manuscript. As well, we have not characterized the presence or heterogeneity of Mn-containing phases or amorphous material in our soils and believe such a characterization would need to proceed a nanoscale investigation of such nodules heterogeneity within our soil systems. As such, while we firmly believe the reviewer's suggestions are highly relevant and would produce interesting and fruitful research, we believe that such work is best pursued in future projects that extend the baseline established herein.